# Novel Bis- and Mono-Pyrrolo[2,3-*d*]pyrimidine and Purine Derivatives: Synthesis, Computational Analysis and Antiproliferative Evaluation

**DOI:** 10.3390/molecules26113334

**Published:** 2021-06-01

**Authors:** Andrea Bistrović Popov, Robert Vianelo, Petra Grbčić, Mirela Sedić, Sandra Kraljević Pavelić, Krešimir Pavelić, Silvana Raić-Malić

**Affiliations:** 1Department of Organic Chemistry, Faculty of Chemical Engineering and Technology, University of Zagreb, Marulićev trg 20, HR-10000 Zagreb, Croatia; ab2602@cam.ac.uk; 2Department of Chemical Engineering and Biotechnology, University of Cambridge, Philippa Fawcet Drive, Cambridge CB3 0AS, UK; 3Division of Organic Chemistry and Biochemistry, Ruđer Bošković Institute, Bijenička 54, HR-10000 Zagreb, Croatia; robert.vianello@irb.hr; 4Department of Biotechnology, University of Rijeka, Ulica Radmile Matejčić 2, HR-51000 Rijeka, Croatia; petra.grbcic@biotech.uniri.hr (P.G.); msedic@biotech.uniri.hr (M.S.); 5Faculty of Health Studies, University of Rijeka, Ulica Viktora Cara Emina 5, HR-51000 Rijeka, Croatia; sandrakp@uniri.hr; 6Faculty of Medicine, Juraj Dobrila University of Pula, HR-52100 Pula, Croatia; pavelic@unipu.hr

**Keywords:** pyrrolo[2,3-*d*]pyrimidines, purine, DFT calculations, ultrasound, antiproliferative activity, pancreatic adenocarcinoma, apoptosis

## Abstract

Novel symmetrical bis-pyrrolo[2,3-*d*]pyrimidines and bis-purines and their monomers were synthesized and evaluated for their antiproliferative activity in human lung adenocarcinoma (A549), cervical carcinoma (HeLa), ductal pancreatic adenocarcinoma (CFPAC-1) and metastatic colorectal adenocarcinoma (SW620) cells. The use of ultrasound irradiation as alternative energy input in Cu(I)-catalyzed azide-alkyne cycloaddition (CuAAC) shortened the reaction time, increased the reaction efficiency and led to the formation of exclusively symmetric bis-heterocycles. DFT calculations showed that triazole formation is exceedingly exergonic and confirmed that the presence of Cu(I) ions is required to overcome high kinetic requirements and allow the reaction to proceed. The influence of various linkers and 6-substituted purine and regioisomeric 7-deazapurine on their cytostatic activity was revealed. Among all the evaluated compounds, the 4-chloropyrrolo[2,3-*d*]pyrimidine monomer **5f** with 4,4′-bis(oxymethylene)biphenyl had the most pronounced, although not selective, growth-inhibitory effect on pancreatic adenocarcinoma (CFPAC-1) cells (IC_50_ = 0.79 µM). Annexin V assay results revealed that its strong growth inhibitory activity against CFPAC-1 cells could be associated with induction of apoptosis and primary necrosis. Further structural optimization of bis-chloropyrrolo[2,3-*d*]pyrimidine with aromatic linker is required to develop novel efficient and non-toxic agent against pancreatic cancer.

## 1. Introduction

Cancer, defined as uncontrolled, rapid and pathological proliferation of cells, is the second leading cause of death, with more than 18 million cases worldwide annually [1]. Pancreatic cancer is predicted to become the second cause of cancer-related deaths by 2030, behind lung cancer [2]. Pancreatic ductal adenocarcinoma is the most common pancreatic cancer type, accounting for more than 90% of cases with a five-year survival rate of less than 9% [3,4]. Current therapy suffers from major limitations due to severe side-effects and multidrug resistance, thereby a continued search to find new and safer anticancer drugs and innovate the development of new cancer treatments are required [5,6]. More than 75% of drugs approved by the FDA and currently available on the market are nitrogen-containing heterocycles due to their ability to easily form hydrogen bonding, dipole-dipole interactions, hydrophobic effects, van der Waals forces and π-stacking interactions with biological targets [7]. The naturally occurring purines play vital roles in numerous life processes [8]. Over the past two decades, purines and their isosteres have appeared as important pharmacophores interacting with the synthesis and functions of nucleic acids and enzymes [9]. To date, 22 pyrimidine-fused bicyclic heterocycles have been approved for clinical use in the treatment of different cancers [10]. Among pyrimidine-fused bicyclic heterocycles, the pyrrolo[2,3-*d*]pyrimidine core can be considered as an isosteric replacement of the biologically relevant purine heterocycle and is hence an important pharmacophore widely used in the field of medicinal chemistry and drug design primarily due to its anticancer properties [11,12,13]. Pyrrolo[2,3-*d*]pyrimidine derivatives exhibited cytotoxic effects in lung and colon cancer cell lines through activation of the mitochondrial apoptotic pathway [14,15,16]. Aryl-substituted pyrrolopyrimidines showed potent inhibition of the membrane bound epidermal growth factor receptor tyrosine kinase (EGFR) and angiogenic inhibitors against human vascular endothelial growth factor receptor-2 (VEGFR-2) that represent important targets in cancer therapy [17,18,19,20,21,22,23,24]. Over the past years, some pyrrolo[2,3-*d*]pyrimidine derivatives were identified as inhibitors of Mer receptor and Src non-receptor tyrosine kinases, isoform of protein kinase B (Akt), mitotic checkpoint kinase (Mps1), Janus kinase 2 (JAK2), and phosphoinositide 3 kinase (PI3K) with promising anticancer activity [25,26,27,28,29,30,31,32,33]. Design strategy in development of purine derivatives as cytostatic agents for kinase inhibition revealed that introduction of cyclic amines improved the activity by forming an additional hydrogen bond to kinase hinge [9,34,35]. In addition, prevalence of halogenated drugs showed that halogen bonds contribute to the stability of protein-ligand complexes [36]. Pyrrolo[2,3-*d*]pyrimidine sulfonamides were recently found to act as cytotoxic agents in hypoxia via inhibition of transmembrane carbonic anhydrases [37]. A 4-(benzylamino)-pyrrolo[2,3-*d*]pyrimidine derivative exerted potent antitumour effects in vivo and induced mitotic cell blockade by impairing both mitotic microtubule organization and dynamics [38]. To overcome multidrug resistance in cancer patients, pyrrolo[2,3-*d*]pyrimidines and purine derivatives with high lipophilicity and molecular weight were developed as potent and selective inhibitors of multidrug resistance-associated protein 1 (MRP1, ABCC1) associated with non-response to chemotherapy in different cancers [39]. Several heterocyclic dimers such as bis-purines [40] and bis-benzimidazoles [41] linked in a head-to-head [42] or head-to-tail manner [43] with acylic and cyclic spacers, have been reported to exhibit anticancer properties by noncovalent interactions with the minor groove of DNA [44].

We previously found that bis-pyrrolo[2,3-*d*]pyrimidine derivatives exhibited potent antiproliferative effect on pancreatic carcinoma (CFPAC-1) cells [45], therefore design strategy led to synthesis of a symmetric series of bis-purines and regioisomeric bis-pyrrolo[2,3-*d*]pyrimidines in which chlorine, amino and cyclic amino groups were introduced at 6-position of the purine and pyrrolo[2,3-*d*]pyrimidine scaffold (Figure 1). Besides, modified aliphatic or π-electron rich aromatic linkers between two heterocycles were introduced to explore further potential for improvement of anti-cancer activity.

To develop a less toxic and more environmentally friendly synthetic method, optimization of the syntheses of target compounds by application of ultrasonic waves and microwaves was performed. Computational analysis was used to elucidate kinetic and thermodynamic aspects of the investigated reactions and identify the precise mechanistic role of the Cu(I) catalyst. Antiproliferative evaluations of bis-purines and bis-pyrrolo[2,3-*d*]pyrimidines and apoptotic mechanism of the selected compound with best antiproliferative effect were also investigated.

## 2. Results and Discussion

### 2.1. Chemistry

The novel bis-pyrrolo[2,3-*d*]pyrimidines(**5a**–**5d**, **6a**–**6d**, **7a**–**7d** and bis-purines **9a**–**9d**, **10a**–**10d**, **11a**–**11d** were synthesized as depicted in Scheme 1 and Scheme 2. *N*-Alkylation of the corresponding pyrrolo[2,3**-***d*]pyrimidine **1a**, more commonly referred to as 7-deazapurine, with 1,2**-**dibromoethane in the presence of K_2_CO_3_ afforded 2**-**bromoethyl derivative **2a**, which were converted to the corresponding 2**-**azidoethyl analogue **3a** using NaN_3_ as previously described [45] (Scheme 1).

Aromatic and aliphatic bis-alkynes: 1,4-bis-(propynyloxy)-benzene (**4a**), 4,4′-bis- (propynyloxy)-1,1′-biphenyl (**4b**), 1,6-heptadiyne (**4c**) and propargyl ether (**4d**) as dipolarophiles, were prepared by *O*-propargylation of the corresponding alcohols with propargyl bromide in the presence of a base [46,47]. Thus, obtained bis-alkynes **4a**−**4d** were subsequently reacted with azide **3a** to give bis-4-chloro-pyrrolo[2,3-*d*]pyrimidines (**5a**−**5d**) and mono-4-chloro-pyrrolo[2,3-*d*]pyrimidines **5f** and **5g**. 4-Chloro-pyrrolo[2,3-*d*]pyrimidine dimers **5a**−**5d** were converted to their 4-piperidine and 4-pyrrolidine- substituted analogues **6a**–**6d** and **7a**–**7d** in very good yields of 60–97% using microwave irradiation [48]. Water, as an environmentally friendly solvent, was used in this reaction which additionally contributed to the implementation of a green and sustainable synthetic approach.

In the purine series, reaction of 2-bromoethyl-6-chloropurine **2b** with sodium azide afforded the diazido derivative **3b** that was found to exist in two tautomeric forms, as reported in the literature [49] and that in CuAAC reaction [50] with bis-alkyne yielded only corresponding monomer **8h** (Scheme 2).

Therefore, the synthetic strategy for the synthesis of 6-substituted bis-purines connected via different spacer was changed. Thus, substitution of 6-chloro to 6-piperidino and 6-pyrrolidino moiety was performed and then CuAAC reaction of heterocyclic azides **3c**−**3e** with bis-alkynes **4a**–**4d** yielded the target 6-substituted bis-purine **9a**–**9d**, **10a**–**10d**, **11a**, **11d**) and mono**-**purine (**9e**–**9g**, **10f**, **11e**, **11g**, **11h**) derivatives (Scheme 2).

In order to optimize the CuAAC reaction, the reactions of bis-alkynes (**4a**−**4d**) and heterocyclic azides (**3a** and **3c**−**3e**) were carried out using different catalysts and reaction conditions to give 6-substituted bis- and mono-purines and 7-deazapurines (Table 1).

Based on the known protocols [51] for CuAAC reaction that include the in situ generation of Cu(I) from a Cu(II) salt or the alternative direct utilization of a Cu(I) source using the combination of CuI/DIPEA/HOAc, which has been found to be a highly efficient catalytic system for this reaction [52], we performed an optimization of the CuAAC reaction using methods A–C.

The most common catalytic system, CuSO_4_ in the presence of metallic copper as a reducing agent (in aqueous *t*-BuOH), was initially chosen in method A. For better performance, we then investigated the application of ultrasound as a green alternative for energy efficient processes in method B. Chemical transformations induced by ultrasound have been described previously [53,54], and it was discovered that ultrasonic irradiation generates a large number of cavitation bubbles, which cause an increase in the local temperature within the reaction mixture and eventually enable the crossing of the activation energy barrier [54]. Finally, copper(I) iodide in the presence of *N*,*N*-diisopropylethylamine (DIPEA) and acetic acid (HOAc) was employed in method C. Comparing the applied synthetic methods A–C in CuAAC reactions of 6**-**chloro**-**7**-**deazapurine azide derivative (**3a**) with all selected bis**-**alkynes, it can be observed that bis**-**triazole dimers **5a**–**5d** were obtained in all methods, while when methods A and C were used with 4,4′**-**bis(propynyloxy)**-**1,1′**-**biphenyl (**4b**) and 1,6**-**heptadiyne (**4c**) the corresponding mono**-**triazole analogues **5f** and **5g** were also obtained (Scheme 1, Table 1).

In the 6-piperidinyl- and 6-pyrrolidinylpurine series, the CuI/DIPEA/HOAc catalytic system using method C was the least selective, yielding both bis- (**9a**–**9d** and **10a**–**10d**) and mono-heterocycles **9e**–**9g** and **10f**. The use of Cu(II) salt as a catalyst in method A afforded only bis-heterocycles in most cases, with exceptions for **5g** and **9e** when small amounts of mono-heterocycles were isolated. However, these reactions were extremely slow and were carried out over 7 days. It can be observed that ultrasound irradiation employed in method B significantly reduced the reaction time to 1.5 h. We may assume that acoustic cavitation in the heterogeneous CuSO_4_/Cu(0) system facilitated mass transfer and surface activation [55] and ultimately accelerated the CuAAC reaction. An additional advantage of the ultrasound-assisted reactions was the exclusive formation of dimeric heterocyclic analogues with improved yields (in the range of 61–82%) compared to reactions without ultrasound irradiation in method A (yields of 28–60%) and method C (yields of 24–64%). In the case of the adenine series, only the ultrasound-assisted reaction (method B) of 2**-**azidoethyladenine **3e** with bis**-**alkyne **4a** afforded bis-adenine **11a** with 1,4-bis(oxymethylene)phenyl linker in low yield (Scheme 2), while **3e** could not react with **4b** in any conditions. When this reaction was performed using CuI, as catalyst, only mono-adenine **11e** was obtained. Also, only CuAAC reactions of azide **3e** with aliphatic bis-alkynes **4c** and **4d** in method C afforded mono-adenine **11g** with propyl chain at C-4 of 1,2,3-triazole, and both bis- **11d** and mono-adenine **11h** in low yields, respectively. Reactions of adenine azide derivative **3e** and bis-alkynes (**4a**–**4d**) did not afford the target products using other applied synthetic methods, indicating the influence of 6-aminopurine on its lower reactivity compared to its 6-piperidino**-** (**3c**) and 6-pyrrolidino**-**substituted (**3d**) purine congeners.

### 2.2. Computational Analysis

Computational analysis was performed to identify precise molecular mechanisms underlying the conversion of azides and alkynes into the matching triazoles, and to reveal the role of the Cu(I) catalysts on the reaction outcomes. For that purpose, we employed a series of DFT calculations on several model systems (Figure 2) with the aim of providing enough structural and electronic features to help interpreting the yields and product distributions observed experimentally.

To demonstrate the necessity to employ the metal catalyst in the CuAAC reaction, we have initially investigated the uncatalyzed conversion between model azide **m2** and neutral model alkyne **m1^0^**, which proceeds in accordance with Figure 3. Bringing reactants into the reactive complex is unfavorable and endergonic by 6.9 kcal mol^−1^, from which it takes additional 21.6 kcal mol^−1^ to reach the transition state corresponding to the concerted formation of both C–N bonds. The latter gives the final triazole in a single step in a very exergonic fashion, linked with the total reaction free energy of Δ*G*_R_ = −51.0 kcal mol^−1^. Still, although thermodynamically favorable, the overall reaction has a rather high kinetic barrier, Δ*G*^‡^ = 28.5 kcal mol^−1^, which renders it as very much unlikely under normal conditions, and emphasizes the necessity to employ a suitable catalyst.

Although alkynes are very weakly acidic systems, and their C–H acidity is typically related with p*K*_a_ values over 20 in water [56], one could still find a suitable base to initiate the deprotonation [57] and convert alkyne **m1^0^** into its anionic form **m1^–^**, with the idea to increase the electrophilicity towards azide and facilitate the reaction. Interestingly, this does not occur (Appendix A), as it takes 7.0 kcal mol^−1^ to form the reactive complex, and, although the kinetic barrier is reduced by 2.6 kcal mol^–1^ to Δ*G*^‡^ = 25.9 kcal mol^−1^, the reaction is thermodynamically significantly less feasible, as the reaction free energy is made less exergonic by 11.6 kcal mol^−1^ to Δ*G*_R_ = −39.4 kcal mol^−1^. The latter will likely prevail and the reaction with deprotonated **m1^–^** will be less favorable. This suggests that attempts to improve the reaction outcomes with strong bases will likely fail.

Introducing Cu(I) ions in the system can result in their complexation with reagents, as both acetylenes and azides are well capable of forming organometallic complexes. Therefore, we proceeded by analyzing potential 1:1 complex with both reagents, which might turn useful in clarifying the role of the metal in the catalysis and identifying which reagent is being activated. The calculated interaction energies (Appendix A) show that Cu(I) ions are much more efficient in binding alkynes (Δ*G*_INT_ = −11.9 kcal mol^−1^ for **m1^0^**) than azides (Δ*G*_INT_ = −5.6 kcal mol^−1^ for **m2**). This is rationalized by electronic distribution as **m1^0^** contains more electronic density within its two carbons (−0.34 |e|) than **m2** within three of its nitrogen atoms (−0.24 |e|), thus the observed trend is predominantly electrostatic in nature.

With this in mind, we analyzed a situation where the cycloaddition reaction occurs with alkyne binding the Cu(I) ion (Figure 4). There, bringing reactants into the reactive complex is exergonic and favorable (−2.8 kcal mol^−1^), while the reaction again proceeds in one step with the kinetic barrier of Δ*G*^‡^ = 19.7 kcal mol^−1^, which represents a considerable 8.8 kcal mol^−1^ reduction from the uncatalyzed reaction. This offers a product complex with Cu(I) binding around alkyne carbons, from which it takes only 1.1 kcal mol^−1^ to detach Cu(I) and allow the final triazole. The obtained reaction profile is highly feasible, while the obtained reduction in the kinetic activation roughly translates into 6–7 orders of magnitude higher rate constant, which is significant and underlines the crucial role of the metal catalyst in facilitating the reaction.

On the other hand, if Cu(I) ions would be able to overcome an initially more favorable placement around alkyne and form a complex with azide prior to reaction (Appendix A), the formation of the reactive complex in that case, would, as anticipated, be less favorable by 5.8 kcal mol^−1^ and even endergonic (+3.0 kcal mol^−1^). In addition to this negative aspect, the obtained product complex featuring triazole with azide-bound Cu(I) is exceedingly exergonic (–64.4 kcal mol^−1^), which will hinder further reaction progress. Specifically, such a stable product complex would increase the energy requirement to finalize the reaction from 1.1 kcal mol^−1^ in the previous case to 13.4 kcal mol^−1^ here. Both of the mentioned aspects will expectedly predominate over formally slightly lower kinetic barrier, by 1.3 kcal mol^−1^ to Δ*G*^‡^ = 18.4 kcal mol^−1^, and make such a conversion less likely. As it was the case with the uncatalyzed reaction, converting alkyne into its deprotonated form **m1^–^** again does not promote the reaction (Appendix A). First, both processes become less exergonic here, with the overall reaction free energy reduced by 10.7 kcal mol^−1^ to Δ*G*_R_ = −40.3 kcal mol^−1^. On top of that, when Cu(I) is bound to the anionic alkyne, the kinetic barrier becomes increased to Δ*G*^‡^ = 19.8 kcal mol^−1^, and even further to Δ*G*^‡^ = 22.0 kcal mol^−1^ when the reaction occurs with the azide-bound metal. All of this again eliminates the necessity to employ any catalytic base in the reaction, let alone that it could, on its own, undergo the complexation reaction with the Cu(I) ions, thus further interfere with the reaction progress in an undesired way.

In summarizing this part, we can emphasize that obtained reaction profiles clearly support the role of the metal catalyst in facilitating the conversion and demonstrate that Cu(I) ions act by binding and activating the alkyne for a successful reaction with nucleophilic azides. Also, the revealed kinetic and thermodynamic aspects seem to suggest that increasing the reaction temperature will likely improve the reaction outcomes, which is found in excellent agreement with a general trend that method B at higher temperatures with ultrasound irradiation offers better reaction yields than method A. Still, to understand different reaction outcomes between methods A and B, where reactive Cu(I) ions are generated in situ, and the method C, where these are directly employed, we must not forget that first two approaches contain both Cu(II) ions and elementary Cu(0) within the solution, which could, at certain cases, impact the reaction and allow different yields, whether higher or lower. To investigate this possibility, we have repeated the analysis considering alternative copper oxidation states.

When Cu(0) is concerned, it more favorably binds to the azide (−20.5 kcal mol^−1^, Appendix A), which even surpasses the most exergonic interaction that charged Cu(I) makes with alkyne by 8.6 kcal mol^−1^, likely being the result of the solvation effect. In that scenario (Appendix A), as expected, the formation of the reactive complex is highly exergonic (−13.7 kcal mol^−1^), yet leading to a very high barrier of Δ*G*^‡^ = 33.0 kcal mol^−1^, which makes this process as unfeasible and renders any impact of Cu(0) on the reaction outcomes as insignificant. We note in passing that a much less stable reactive complex involving the alkyne-bound Cu(0) would proceed in a stepwise fashion with both C–N bonds formed separately (Appendix A) and the rate-limiting second step. On the other hand, Cu(II) ions reveal a very interesting trend as their ability to complex reactants is precisely identical for both alkyne and azide at Δ*G*_R_ = −13.0 kcal mol^−1^ (Appendix A), which makes both options viable. However, the reactive complex with the alkyne-bound Cu(II) is by 6.2 kcal mol^−1^ more favorable, which directs the reactivity towards the stepwise triazole formation (Appendix A), where both N–C bonds are created under similar kinetic requirements. Still, the process to form the second bond is a bit more demanding and defines the rate-limiting step, yet the overall activation energy is only Δ*G*^‡^ = 8.3 kcal mol^−1^, which hints at a possible impact of Cu(II) ions on the overall conversion. However, one must emphasize that this reaction is again associated with a rather high energy cost of 12.7 kcal mol^−1^ to move from the reactive complex and detach Cu(II) to allow the final triazole, which might hinder any positive effect, let alone the availability of Cu(II) to undertake the reaction in the first place. Which of these aspects will prevail in solution and whether the impact of Cu(II) will be significant or insignificant at all, is difficult to say, and may likely depend on a particular reaction condition and different electronic and structural features of the employed reactants. As such, using Cu(II) salts in a combination with elementary copper, in order to generate Cu(I) in situ, may and likely will generate slightly different conditions for the alkyne-azide cycloaddition reaction than when Cu(I) salts are directly used, which justifies why different trends are experimentally detected and no wide-ranging conclusions can be made in this respect.

Lastly, a somewhat general trend emerging from experiments is that 6-chloro derivatives are typically more reactive than their 6-amino analogues when reaction times or yields are compared. At first, such differences appear surprising given a large distance of 6-substituents from the reacting triazole to exhibit any direct impact, or the fact that triazole is separated from the aromatic fragment by two methylene units for any indirect influence of these substituents through resonance/inductive effects. Therefore, a plausible explanation for the observed reactivity differences could be linked with their abilities to form complexes with metal catalysts (Appendix A). Namely, electron-withdrawing chlorine depletes the electron density from the aromatic skeleton, which diminishes the complex formation with Cu(I). In contrast, electron-donating amines consistently increase the tendency to complex Cu(I) in solution in both pyrimidine and purine derivatives, which could reduce the catalytic efficiency of the metal, thus somewhat lower reactivity of 6-amino derivatives.

### 2.3. Biological Evaluations

#### 2.3.1. Antiproliferative Evaluations

The antiproliferative results obtained for the prepared compounds on human tumor cell lines, including lung adenocarcinoma (A549), cervical carcinoma (HeLa), ductal pancreatic adenocarcinoma (CFPAC-1) and metastatic colorectal adenocarcinoma (SW620) cells, are presented in Table 2. Compounds that exhibited marked cytostatic activity (**BIS-PP2**, **5b**, **5f**, **7a**, **9a**, **9b**, **9f**, **10b** and **10f**) were additionally evaluated for their cytotoxic effects on normal human foreskin fibroblasts (HFF-1), as indicated in the footnote of the Table 2. For comparison, anti-proliferative effects of chloropyrrolo[2,3-*d*]pyrimidine structural analogues BIS-PP1, BIS-PP2 [45] were also included in Table 2.

From the series of 4-chloropyrrolo[2,3-*d*]pyrimidines, compound **5f,** with the aromatic unit, showed the most potent cytostatic activity, particularly on HeLa (IC_50_ = 0.98 µM) and CFPAC-1 (IC_50_ = 0.79 µM) cells. Antiproliferative effects of **5f** were somewhat better than those of previously published structural analogues **BIS-PP1**, **BIS-PP2** [45]. Importantly, this compound showed to be more selective than BIS-PP1, BIS-PP2 exhibiting lower toxicity to normal fibroblasts (HFF-1). On the contrary, compounds **5c**, **5d** and **5g** with aliphatic propylene and oxydimethylene central unit exhibited only moderate activity on HeLa, CFPAC-1 and SW620 cell lines.

Among cyclic amino-substituted bis-pyrrolo[2,3-*d*]pyrimidines, 4-pyrrolidine analogue **7a** with 1,4-bis(oxymethylene)phenyl linker exhibited marked growth-inhibitory effect (IC_50_ = 6.4 µM) on HeLa cells, while 4-piperidine analogue **6b** had strong activity (IC_50_ = 17.4 µM) on CFPAC-1 cells. Only 4-piperidine pyrrolo[2,3-*d*]pyrimidines **6c** and **6d**, with aliphatic spacers, had moderate antiproliferative activities in the range of 28.2−65.2 µM compared to corresponding analogues **5c**, **5d** and **7c**, **7d**.

In the series of bis- and mono-purines, 6-piperidinyl- (**9b** and **9f**) and 6-pyrrolidinylpurines (**10b** and **10f**) containing 4,4′-bis(oxymethylene)biphenyl exhibited marked, although rather non-specific inhibitory effects. Bis-purines **9b** and **10b** showed the highest activity on HeLa cell line (**9b**: IC_50_ = 3.8 µM; **10b**: IC_50_ = 7.4 µM). Structural analogues **9a**, **9e** and **10a** with 1,4-bis(oxymethylene)phenyl linker displayed decreased activity, while compounds with aliphatic unit (**9c**, **9g**, **9d** and **10c**, **10d**) showed only moderate to marginal growth-inhibition, that is in accordance with the findings for pyrrolo[2,3-*d*]pyrimidines. Both bis- and mono-adenine derivatives showed no inhibitory activity (IC_50_ > 100 µM) on all evaluated tumor cell lines.

Comparing to **BIS-PP2**, it can be observed that structural modifications in novel bis- and mono-pyrrolo[2,3-*d*]pyrimidine and purine derivatives reduced their cytotoxic effects on normal HFF-1 cells to a lesser extent. Compounds with best antiproliferative activities also exhibited inhibitory effects on normal HFF-1 cells.

Overall, it can be observed that the linker between the heterocycle scaffolds in the symmetrical 7-deazapurines and bis-purines had a significant impact on antitumor activity (Figure 5).

Compounds with aliphatic linkers (**5c**–**7c**, **5d**–**7d**, **9c**, **10c** and **9d**–**11d**) exhibited moderate activity or were deprived of any antitumor activities, while introducing the 4,4′-bis(oxymethylene)biphenyl linker (**5b**, **9b** and **10b**) caused an enhancement of tumor cell growth inhibition. The influence of heterocyclic scaffold was also observed showing that series of cyclic 6-amino bis-purines (**9a**−**9d** and **10a**−**10d**) were generally more active than the corresponding cyclic 4-amino bis-pyrrolo[2,3-*d*]pyrimidine (**6a**−**6d** and **7a**−**7d**) analogues. Comparison of the antiproliferative activity of all purine derivatives showed that cyclic amines in purine derivatives improved the activity relative to adenine derivatives that did not exhibit inhibitory activities.

#### 2.3.2. Apoptosis Detection

Annexin V assay was performed to determine if the antiproliferative activity of compound **5f** exhibiting the most pronounced potency in ductal adenocarcinoma cancer cell line (CFPAC-1) could be attributed to induction of apoptosis. Obtained data are presented in Table 3 and Figure 6.

Table 3 shows percentages of cells in different stages of apoptosis. It can be observed that compound **5f** showed pro-apoptotic effect in CFPAC-1 cells as early as 48 h after treatment with both 2 × IC_50_ (1.58 µM) and 5 × IC_50_ (3.95 µM) concentrations. Following 48-h treatment, both concentrations of compound **5f** showed pro-apoptotic activity, where 5 × IC_50_ concentration induced decrease in cell viability by 28.4% and increase in the percentage of cells that underwent early apoptosis by 21.21%. After 72 h of treatment, a significant decrease in cell viability was noticed in both treatments concomitantly with a profound increase in early apoptotic cells after 2 × IC_50_ and 5 × IC_50_ treatments by 24.08% and 33.35%, respectively, followed by a dramatic rise in the late apoptotic/primary necrotic cells by 26.14% that occurred after 5 × IC_50_ treatment. Collectively, these results show that compound **5f** induces apoptosis and primary necrosis in CFPAC-1 cells in a concentration- and time-dependent manner.

## 3. Materials and Methods

### 3.1. General

All the solvents and chemicals were purchased from Aldrich (St. Louis, MO, USA) and Acros (Geel, Belgium). Thin layer chromatography was performed on pre-coated silica gel 60F-254 plates (Merck, Kenilworth, NJ, USA ) while glass column slurry-packed under gravity with silica gel (0.063–0.2 mm Fluka, Seelze, Germany) was employed for column chromatography. Melting points of compounds were determined using a Kofler micro hot stage. ^1^H- and ^13^C-NMR spectra were recorded on a Bruker 300 and 600 MHz spectrometers (Bruker, Billerica, MA, USA). All data were recorded in dimethyl sulfoxide (DMSO-*d*_6_) at 298 K. Chemical shifts were referenced to the residual solvent signal of DMSO at *δ* 2.50 ppm for ^1^H and *δ* 39.50 ppm for ^13^C. Individual resonances were assigned based on their chemical shifts, signal intensities, multiplicity of resonances, H–H coupling constants. The ultrasound-assisted reactions were carried out in a Bandelin Bath Cleaner (Sonorex Digital 10 P, Berlin, Germany) with a nominal power of 1000 W and frequency of 35 kHz. The reactions were carried out in a round-bottomed flask of 25 mL capacity suspended at the center of the cleaning bath, 5 cm below the surface of the liquid. Microwave-assisted syntheses were performed in a Milestone start S microwave oven) (Sorisole, Italy) using glass cuvettes at 100 °C and 400 W.

### 3.2. Experimental Procedure for the Synthesis of Compounds

The compounds 6-(piperidin-1-yl)-9*H*-purine (**1c**) [58], 6-(pyrrolidin-1-yl)-9*H*-purine (**1d**) [59], 7-(2-bromoethyl)-4-chloro-7*H*-pyrrolo[2,3-*d*]pyrimidine (**2a**) [45], 9-(2-bromoethyl)-9*H*-purin-6-amine (**2e**) [60], 7-(2-azidoethyl)-4-chloro-7*H*-pyrrolo[2,3-*d*]pyrimidine (**3a**) [45], 9-(2-azidoethyl)-9*H*-purin-6-amine (**3e**) [60], 1,4-bis(prop-2-yn-1-yloxy)benzene (**4a**) [46] and 4,4′-bis(prop-2-yn-1-yloxy)-1,1′-biphenyl (**4b**) [47] were prepared according to known procedures.

### 3.3. General Procedure for the N-Alkylation of Compounds

The corresponding heterocyclic base **1a**–**1e** was dissolved in dry DMF (8 mL), K_2_CO_3_ was added (1.2 eq) and the reaction mixture was stirred for 1 h. 1,2-Dibromoethane (1.2 eq) was added to the mixture and stirred for 24 h at room temperature. The solvent was evaporated to dryness and the residue was purified by column chromatography.

*9-(2-Bromoethyl)-6-chloro-9H-purine* (**2b**)**.** Compound **2b** was prepared according to the abovementioned procedure from compound **1b** (1 g, 7.40 mmol). After purification by column chromatography (CH_2_Cl_2_:MeOH = 50:1) compound **2b** was obtained as white crystal (1.24 g, 74%, m.p. = 110–112 °C). ^1^H-NMR (300 MHz, DMSO*-d*_6_) (*δ*/ppm): 8.81 (1H, s, H2), 8.76 (1H, s, H8), 4.74 (2H, t, *J* = 6.0 Hz, CH_2_CH_2_), 4.01 (2H, t, *J* = 6.0 Hz, CH_2_CH_2_). ^13^C-NMR (75 MHz, DMSO*-d*_6_) (*δ*/ppm): 151.9 (C6), 151.6 (C2), 149.1 (C4), 147.5 (C8), 130.8 (C5), 45.3 (CH_2_), 31.1 (CH_2_).

*9-(2-Bromoethyl)-6-(piperidine-1-yl)-9H-purine* (**2c**)**.** Compound **2c** was prepared according to the abovementioned procedure from compound **1c** (550 mg, 2.71 mmol). After purification by column chromatography (CH_2_Cl_2_:MeOH = 100:1) compound **2c** was obtained as white solid (650 mg, 76%, m.p. = 148–150 °C). ^1^H-NMR (300 MHz, DMSO*-d*_6_) (*δ*/ppm): 8.22 (1H, s, H2), 8.19 (1H, s, H8), 4.57 (2H, t, *J* = 6.1 Hz, CH_2_CH_2_), 4.19 (4H, bs, CH_2_), 3.94 (2H, t, *J* = 6.1 Hz, CH_2_CH_2_), 1.78–1.42 (6H, m, CH_2_). ^13^C-NMR (75 MHz, DMSO*-d*_6_) (*δ*/ppm): 153.1 (C6), 151.9 (C2), 150.5 (C4), 139.7 (C8), 118.8 (C5), 45.6 (CH_2_), 44.6 (CH_2_), 31.4 (CH_2_), 25.6 (CH_2_), 24.2 (CH_2_).

*9-(2-Bromoethyl)-6-(pyrrolidine-1-yl)-9H-purin* (**2d**) Compound **2d** was prepared according to the abovementioned procedure from compound **1d** (650 mg, 3.44 mmol). After purification by column chromatography (CH_2_Cl_2_:MeOH = 100:1) compound **2d** was obtained as white solid (650 mg, 64%, m.p. = 162–164 °C). ^1^H-NMR (300 MHz, DMSO*-d*_6_) (*δ*/ppm): 8.21 (1H, s, H2), 8.16 (1H, s, H8), 4.57 (2H, t, *J* = 6.0 Hz, CH_2_CH_2_), 4.04 (2H, s, CH_2_), 3.94 (2H, t, *J* = 6.1 Hz, CH_2_CH_2_), 3.63 (2H, s, CH_2_), 1.95 (6H, s, CH_2_). ^13^C-NMR (75 MHz, DMSO*-d*_6_) (*δ*/ppm): 150.7 (C2), 149.4 (C4), 140.9 (C8), 119.2 (C5), 59.2 (CH_2_), 45.9 (CH_2_), 44.7 (CH_2_), 31.5 (CH_2_).

### 3.4. General Procedure for the Synthesis of Azidoethyl Derivatives

The corresponding 2-bromoethyl derivative **2a**–**2f** was dissolved in acetone. NaN_3_ (4 eq) dissolved in water (~ 3 mL) was added dropwise to the reaction mixture and stirred under reflux overnight. The solvent was evaporated to dryness and the residue dissolved in ethyl-acete (60 mL) and extracted with brine. The organic layer was dried over MgSO_4_, filtered and evaporated.

*6-Azido-9-(2-azidoethyl)-9H-purine* (**3b**). Compound **3b** was prepared according to the abovementioned procedure from compound **2b** (652 mg, 2.49 mmol) to give compound **3b** as white solid (230.2 mg. 63%; m.p. = 108–112 °C). ^1^H-NMR (600 MHz, DMSO*-d*_6_) (*δ*/ppm): 10.13 (1H, s, H2), 8.70 (1H, s, H8), 4.62 (2H, t, *J* = 5.6 Hz, CH_2_CH_2_), 3.93 (2H, t, *J* = 5.6 Hz, CH_2_CH_2_). ^13^C-NMR (75 MHz, DMSO*-d*_6_) (*δ*/ppm): 145.4 (C6), 144.5 (C2), 142.4 (C4), 135.9 (C8), 119.7 (C5), 49.9 (CH_2_), 43.6 (CH_2_).

*9-(2-Azidoethyl)-6-(piperidin-1-yl)-9H-purine* (**3c**). Compound **3c** was prepared according to the abovementioned procedure from compound **2c** (650 mg, 2.10 mmol) to give compound **3c** as white solid (566.3 mg, 99%, m.p. = 78–80 °C). ^1^H-NMR (300 MHz, DMSO*-d*_6_) (*δ*/ppm): 8.22 (1H, s, H2), 8.18 (1H, s, H8), 4.35 (2H, t, *J* = 5.7 Hz, CH_2_CH_2_), 4.19 (4H, s, CH_2_), 3.81 (2H, t, *J* = 5.7 Hz, CH_2_CH_2_), 1.72–1.52 (6H, m, CH_2_CH_2_). ^13^C-NMR (75 MHz, DMSO*-d*_6_) (*δ*/ppm): 153.1 (C6), 151.9 (C2), 150.6 (C4), 139.7 (C8), 118.9 (C5), 49.6 (CH_2_), 42.4 (CH_2_), 25.7 (CH_2_), 24.2 (CH_2_).

*9-(2-Azidoethyl)-6-(pyrrolidin-1-yl)-9H-purine* (**3d**). Compound **3d** was prepared according to the abovementioned procedure from compound **2e** (400 mg, 1.35 mmol) to give compound **3d** as white solid (345.1 mg, 99%, m.p. = 96–101 °C). ^1^H-NMR (300 MHz, DMSO*-d*_6_) (*δ*/ppm): 8.21 (1H, s, H2), 8.15 (1H, s, H8), 4.35 (2H, t, *J* = 5.7 Hz, CH_2_CH_2_), 4.05 (2H, bs, CH_2_), 3.80 (2H, t, *J* = 5.7 Hz, CH_2_CH_2_), 3.63 (2H, bs, CH_2_), 1.95 (4H, bs, CH_2_). ^13^C-NMR (151 MHz, DMSO*-d*_6_) (*δ*/ppm): 152.5 (C6), 152.2 (C2), 150.0 (C4), 140.2 (C8), 119.3 (C5), 49.7 (CH_2_), 42.3 (CH_2_).

### 3.5. General Procedure for the Synthesis of bis- (**5a**–**5d**, **9a**–**9d**, **10a**–**10d**, **11a**, **11d**) and Mono- (**5f**, **5g**, **9e**–**9g**, **10f**, **11e**, **11g**, **11h**) Pyrrolo[2,3-d]pyrimidines and Purines

Method A: The reaction mixture of the corresponding bis-alkyne **4a**–**4d** (1 eq), 2-azidoethyl base **3a**–**3e** (2.2 eq), Cu(0) (1 eq) and 1M CuSO_4_ (0.3 eq) in 1 mL DMF and a mixture of *t*-BuOH:H_2_O = 1: 1 (3 mL) was stirred at room temperature for 120 h. The solvent was evaporated and the residue was purified by column chromatography using CH_2_Cl_2_, as an initial eluent, and CH_2_Cl_2_:CH_3_OH = 10:1, as final eluent.

Method B: The reaction mixture of the corresponding bis-alkyne **4a**–**4d** (1 eq), 2-azidoethyl base **3a**–**3e** (2.2 eq), Cu(0) (1 eq) and 1M CuSO_4_ (0.3 eq) in 1 mL DMF and a mixture of *t*-BuOH:H_2_O = 1:1 (3 mL) was placed in an ultrasonic bath cleaner (1000 W, 35 kHz) at 80 °C for 1.5 h. The solvent was evaporated and the residue was purified by column chromatography using CH_2_Cl_2_, as an initial eluent, and CH_2_Cl_2_:CH_3_OH = 10:1, as final eluent.

Method C: The reaction mixture of the corresponding bis-alkyne **4a**–**4d** (1 eq), 2-azidoethyl base **3a**–**3e** (2.2 eq), CuI(0) (0.2 eq), *N,N*-diisopropylethylamine (4 eq) and acetic acid (4 eq) in CH_2_Cl_2_ (1 mL) was stirred at room temperature for 72 h. The solvent was evaporated and the residue was purified by column chromatography using CH_2_Cl_2_, as an initial eluent, and CH_2_Cl_2_:CH_3_OH = 10:1, as final eluent.

*1,4-Bis{[1-(2-(4-chloro-7H-pyrrolo[2,3-d]pyrimidin-7-yl)ethyl)-1H-1,2,3-triazol-4-yl]methoxy}benzene* (**5a**). Compound **5a** was prepared according to the abovementioned procedure from compound **4a** (method A: 50 mg, 0.27 mmol; method B: 22 mg, 0.11 mmol; method C: 50 mg, 0.27 mmol) and compound 3a (method A: 132.2 mg, 0.59 mol; method B: 58 mg, 0.44 mmol; method C: 123.2 mg, 0.59 mmol) to obtain **5a** as white solid (method A: 83.5 mg, 49%; method B: 50 mg, 67%; method C: 40.4 mg, 24%; m.p. = 234–239 °C). ^1^H-NMR (300 MHz, DMSO*-d*_6_) (*δ*/ppm): 8.54 (2H, s, H2), 8.01 (2H, s, H5′), 7.49 (2H, d, *J* = 3.6 Hz, H6), 6.89 (4H, s, Ph), 6.57 (2H, d, *J* = 3.6 Hz, H5), 4.98 (4H, s, CH_2_), 4.89–4.83 (4H, m, CH_2_CH_2_), 4.79–4.73 (4H, m, CH_2_CH_2_). ^13^C-NMR (151 MHz, DMSO*-d*_6_) (*δ*/ppm): 152.2 (Ph-q), 150.7 (C4), 150.5 (C7a), 150.3 (C2), 143.0 (C4′), 131.1 (C6), 124.7 (C5′), 116.7 (C4a), 115.6 (Ph), 98.8 (C5), 61.4 (OCH_2_), 49.0 (CH_2_CH_2_), 44.7 (CH_2_CH_2_). Anal. calcd. for C_28_H_24_Cl_2_N_12_O_2_: C, 53.26; H, 3.83; N, 26.62. Found: C, 53.56; H, 3.95; N, 26.44.

*4,4′-Bis{[1-(2-(4-chloro-7H-pyrrolo[2,3-d]pyrimidin-7-yl)ethyl)-1H-1,2,3-triazol-4-yl]-methoxy}-1,1′-biphenyl* (**5b**) and *4-chloro-7-{2-[4-(((4′-(prop-2-yn-1-yloxy)-[1,1′-biphenyl]-4-yl)oxy)methyl)-1H-1,2,3-triazol-1-yl]ethyl}-7H-pyrrolo[2,3-d]pyrimidine* (**5f**). Compounds **5b** and **5f** were prepared according to the abovementioned procedure from compound **4b** (method A: 50 mg, 0.19 mmol; method B: 29.4 mg, 0.11 mmol; method C: 50 mg; 0.19 mmol) and compound **3a** (method A: 93.1 mg, 0.42 mol; method B: 55 mg, 0.24 mmol; method C: 93.1 mg, 0.42 mmol) to obtain **5b** as white solid (method A: 46.4 mg, 34%; method B: 94.4 mg, 66%; method C: 26.2 mg, 35%; m.p. > 260 °C) and **5f** as yellow solid (method C: 8 mg, 14%; m.p. = 234–238 °C).

**5b**: ^1^H-NMR (600 MHz, DMSO*-d*_6_) (*δ*/ppm): 8.54 (2H, s, H2), 8.06 (2H, s, H5′), 7.54 (4H, d, *J* = 8.7 Hz, Ph), 7.49 (2H, d, *J* = 3.6 Hz, H6), 7.03 (4H, d, *J* = 8.7 Hz, Ph), 6.56 (2H, d, *J* = 3.6 Hz, H5), 5.08 (4H, s, OCH_2_), 4.90–4.86 (4H, m, CH_2_CH_2_), 4.79–4.75 (4H, m, CH_2_CH_2_). ^13^C-NMR (75 MHz, DMSO*-d*_6_) (*δ*/ppm): 157.1 (Ph-q), 150.7 (C4), 150.5 (C7a), 150.2 (C2), 142.8 (C4′), 132.5 (Ph-q), 131.1 (C5), 127.2 (Ph), 124.8 (C5′), 116.7 (C4a), 115.1 (Ph), 98.7 (C6), 61.0 (OCH_2_), 49.0 (CH_2_CH_2_), 44.6 (CH_2_CH_2_). Anal. calcd. for C_34_H_28_Cl_2_N_12_O_2_: C, 57.71; H, 3.99; N, 23.75. Found: C, 57.65; H, 3.73; N, 23.74.

**5f**: ^1^H-NMR (300 MHz, DMSO*-d*_6_) (*δ*/ppm): 8.54 (1H, s, H2), 8.05 (1H, s, H5′), 7.62–7.49 (4H, m, Ph), 7.49 (1H, d, *J* = 3.6 Hz, H6), 7.16–6.92 (4H, m, Ph), 6.57 (1H, d, *J* = 3.6 Hz, H5), 5.08 (2H, s, OCH_2_), 4.91–4.84 (2H, m, *J* = 5.4 Hz, CH_2_CH_2_), 4.82 (2H, d, *J* = 2.3 Hz, CH_2_CCH), 4.80–4.74 (2H, m, CH_2_CH_2_), 3.58 (1H, t, *J* = 2.3 Hz, CH_2_CCH). ^13^C-NMR (75 MHz, DMSO*-d*_6_) (*δ*/ppm): 157.1 (Ph-q), 150.5 (C7a), 150.2 (C2), 142.8 (C4′), 132.5 (Ph-q), 131.1 (Ph), 127.2 (Ph), 124.7 (C5′), 116.7 (C4a), 115.1 (Ph/C5), 98.8 (C6), 79.4 (CCH), 61.0 (OCH_2_), 49.0 (CH_2_CH_2_), 45.7 (CH_2_CH), 44.6 (CH_2_CH_2_). Anal. calcd. for C_26_H_21_ClN_6_O_2_: C, 64.40; H, 4.37; N, 17.33. Found: C, 64.56; H, 4.33; N, 17.42.

1,3-Bis{1-[2-(4-chloro-7H-pyrrolo[2,3-d]pyrimidin-7-yl)ethyl]-1H-1,2,3-triazol-4-yl}propane (**5c**) and 4-chloro-7-{2-[4-(pent-4-yn-1-yl)-1H-1,2,3-triazol-1-yl]ethyl}-7H-pyrrolo[2,3-d]pyrimidine (**5g**). Compounds **5c** and **5g** were prepared according to the abovementioned procedure from 1,6-heptadiyne **4c** (method A: 10.5 mg, 0.11 mmol; method B: 100 mg, 1.09 mmol; method C: 50 mg, 0.54 mmol) and compound **3a** (method A: 56 mg, 0.25 mmol; method B: 534.3 mg, 2.4 mol; method C: 264.5 mg, 1.19 mmol) to obtain **5c** as white solid (method A: 26.6 mg, 45%; method B: 258 mg, 62%; method C: 183 mg, 63%; m.p. = 208–210 °C) and **5g** as yellow oil (method A: 4 mg, 11%; method C: 51.5 mg, 35%).

**5c**: ^1^H-NMR (300 MHz, DMSO*-d*_6_) (*δ*/ppm): 8.54 (2H, s, H2), 7.62 (2H, s, H5′), 7.52 (2H, d, *J* = 3.6 Hz, H6), 6.59 (2H, d, *J* = 3.6 Hz, H5), 4.87–4.67 (8H, m, CH_2_CH_2_), 2.43 (4H, t, *J* = 7.4 Hz, CH_2_CH_2_CH_2_), 1.75–1.57 (2H, m, CH_2_CH_2_CH_2_). ^13^C-NMR (75 MHz, DMSO*-d*_6_) (*δ*/ppm): 150.7 (C4), 150.5 (C7a), 150.2 (C2), 146.3 (C4’), 131.1 (C6), 122.1 (C5’), 116.7 (C4a), 98.7 (C5), 48.8 (CH_2_CH_2_), 45.0 (CH_2_CH_2_), 28.7 (CH_2_), 24.0 (CH_2_). Anal. calcd. for C_23_H_22_Cl_2_N_12_: C, 51.40; H, 4.13; N, 31.28. Found: C, 51.66; H, 4.09; N, 31.35.

**5g**: ^1^H-NMR (300 MHz, DMSO*-d*_6_) (*δ*/ppm): 8.55 (1H, s, H2), 7.65 (1H, s, H5′), 7.52 (1H, d, *J* = 3.6 Hz, H6), 6.60 (1H, d, *J* = 3.6 Hz, H5), 4.97–4.64 (4H, m, CH_2_CH_2_), 2.78 (1H, t, *J* = 2.6 Hz, CH), 2.58 (2H, t, *J* = 7.4 Hz, CH_2_), 2.08 (2H, td, *J* = 7.1, 2.6 Hz, CH_2_), 1.71–1.53 (2H, m, CH_2_). ^13^C-NMR(151 MHz, DMSO*-d*_6_) (*δ*/ppm): 150.8 (C4), 150.6 (C7a), 150.2 (C2), 146.1 (C4′), 131.1 (C6), 122.3 (C5′), 116.7 (C4a), 98.8 (C5), 84.1 (CCH), 71.4 (CCH), 48.9 (CH_2_CH_2_), 44.7 (CH_2_CH_2_), 27.9 (CH_2_), 23.8 (CH_2_), 17.0 (CH_2_). Anal. calcd. for C_15_H_15_ClN_6_: C, 57.24; H, 4.80; N, 26.70. Found: C, 57.31; H, 4.90; N, 26.57.

*{1-[2-(4-Chloro-7H-pyrrolo[2,3-d]pyrimidin-7-yl)ethyl]-1H-1,2,3-triazol-4-yl}methyl ether* (**5d**). Compound **5d** was prepared according to the abovementioned procedure from propargyl ether **4d** (100 mg, 1.06 mmol) and compound **3a** (495.6 mg, 2.33 mmol) to obtain **5d** as white solid (method A: 290.1 mg, 51%; method C: 363 mg, 64%; m.p. = 206–208 °C). ^1^H-NMR (300 MHz, DMSO*-d*_6_) (*δ*/ppm): 8.54 (2H, s, H2), 7.93 (2H, s, H5′), 7.52 (2H, d, *J* = 3.6 Hz, H6), 6.59 (2H, d, *J* = 3.6 Hz, H5), 4.89–4.83 (4H, m, CH_2_CH_2_), 4.80–4.74 (4H, m, CH_2_CH_2_), 4.34 (4H, s, CH_2_OCH_2_). ^13^C-NMR (75 MHz, DMSO*-d*_6_) (*δ*/ppm): 150.7 (C4), 150.5 (C7a), 150.2 (C2), 143.5 (C4′), 131.1 (C6), 124.3 (C5′), 116.7 (C4a), 98.8 (C5), 62.0 (CH_2_OCH_2_), 48.9 (CH_2_CH_2_), 44.6 (CH_2_CH_2_). Anal. calcd. for C_22_H_20_Cl_2_N_12_O: C, 48.99; H, 3.74; N, 31.16. Found: C, 49.01; H, 3.62; N, 31.16.

*7-{2-[4-((Prop-2-yn-1-yloxy)methyl)-1H-1,2,3-triazol-1-yl]ethyl}-7H-tetrazolo[5,1-i]purine* (**8h**) Compound **8h** was prepared according to the abovementioned procedure for method B from propargyl ether **4d** (0.10 mL, 1.01 mmol) and compound **3b** (513.1 mg, 2.23mmol) to obtain **8h** as white crytals (189.4 mg, 59%; m.p. = 149–148 °C). ^1^H (300 MHz, DMSO*-d*_6_) (*δ*/ppm): 10.07 (1H, s, H2), 8.39 (1H, s, 8H), 7.95 (1H, s, H5′), 4.99–4.87 (4H, m, CH_2_CH_2_), 4.46 (2H, s, CH_2_), 4.06 (2H, d, *J* = 2.4 Hz, CH_2_CCH), 3.44 (1H, t, *J* = 2.4 Hz, CH_2_CCH). ^13^C (75 MHz, DMSO*-d*_6_) (*δ*/ppm): 145.4 (C6), 143.3 (C4), 142.3 (C4′), 135.7 (C8), 124.6 (C5′), 119.6 (C5), 79.9 (CCH), 77.4 (CCH), 61.7 (CH_2_), 56.4 (CH_2_), 48.9 (CH_2_CH_2_), 44.5 (CH_2_CH_2_). Anal. calcd. for C_13_H_12_N_10_O: C, 48.15; H, 3.73; N, 43.19. Found: C, 48.08; H, 3.76; N, 43.29.

1,4-Bis{[1-(2-(6-(piperidin-1-yl)-9H-purin-9-yl)ethyl)-1H-1,2,3-triazol-4-yl]methoxy}benzene (**9a**) and 6-(piperidin-1-yl)-9-{2-[4-((4-(prop-2-yn-1-yloxy)phenoxy)methyl)-1H-1,2,3-triazol-1-yl]ethyl}-9H-purine (**9e**) Compounds **9a** and **9e** were prepared according to the above-mentioned procedure from compound **4a** (50 mg, 0.27 mmol) and compound **3c** (160.7 mg, 0.59 mmol) to obtain **9a** as white solid (method A: 58.1 mg, 28%; method B: 121.1 mg, 61%; method C: 67.5 mg, 36%; m.p. = 233–236 °C) and **9e** (method A: 4.9 mg, 4%; method C: 34.4 mg, 29%; m.p. = 158–160 °C).

**9a**: ^1^H-NMR (300 MHz, DMSO*-d*_6_) (*δ*/ppm): 8.18 (2H, s, H2), 8.12 (2H, s, H8), 7.83 (2H, s, H5′), 6.90 (4H, s, Ph), 5.01 (4H, s, CH_2_), 4.87 (4H, t, *J* = 5.7 Hz, CH_2_CH_2_), 4.65 (4H, t, *J* = 5.7 Hz, CH_2_CH_2_), 4.16 (8H, bs, CH_2_), 1.70–1.51 (12H, m, CH_2_). ^13^C-NMR (75 MHz, DMSO*-d*_6_) (*δ*/ppm): 153.2 (C6), 152.4 (Ph-q), 152.1 (C2), 150.6 (C4), 143.3 (C4′), 139.5 (C8), 124.8 (C5′), 118.9 (C5), 115.9 (Ph), 61.7 (CH_2_), 48.8 (CH_2_), 43.2 (CH_2_), 25.8 (CH_2_), 24.4 (CH_2_). Anal. calcd. for C_36_H_42_N_16_O_2_: C, 59.16; H, 5.79; N, 30.66. Found: C, 59.36; H, 5.64; N, 30.62.

**9e**: ^1^H-NMR (300 MHz, DMSO*-d*_6_) (*δ*/ppm): 8.19 (1H, s, H2), 8.12 (1H, s, H8), 7.83 (1H, s, H5′), 6.92 (4H, d, *J* = 1.9 Hz, Ph), 5.02 (2H, s, CH_2_), 4.87 (2H, t, *J* = 5.7 Hz, CH_2_CH_2_), 4,71 (2H, d, *J* = 2.4 Hz, CH_2_CCH), 4.66 (2H, t, *J* = 5.7 Hz, CH_2_CH_2_), 4.17 (4H, s, CH_2_), 3.53 (1H, t, *J* = 2.4 Hz, CH_2_CCH), 1.74–1.48 (6H, m, CH_2_). ^13^C-NMR (75 MHz, DMSO*-d*_6_) (*δ*/ppm): 153.0 (C6), 152.6 (Ph-q), 151.9 (C2), 151.4 (Ph-q), 150.5 (C4), 143.0 (C4′), 138.5 (C8), 124.6 (C5′), 118.7 (C5), 115.8 (Ph), 115.6 (Ph), 78.0 (CCH), 61.5 (CH_2_), 55.9 (CH_2_), 48.5 (CH_2_), 43.0 (CH_2_), 25.6 (CH_2_), 24.2 (CH_2_). Anal. calcd. for C_24_H_26_N_8_O_2_: C, 62.87; H, 5.72; N, 24.44. Found: C, 62.91; H, 5.77; N, 24.28.

4,4′-Bis((1-(2-(6-(piperidin-1-yl)-9H-purin-9-yl)ethyl)-1H-1,2,3-triazol-4-yl)methoxy)-1,1′-biphenyl (**9b**) and 6-(piperidin-1-yl)-9-{2-[4-(((4′-(prop-2-yn-1-yloxy)-[1,1′-biphenyl]-4-yl)oxy)methyl)-1H-1,2,3-triazol-1-yl]ethyl}-9H-purine (**9f**). Compounds **9b** and **9f** were prepared according to the abovementioned procedure from compound **4b** (50 mg, 0.19 mmol) and compound **3c** (113.8 mg, 0.42 mmol) to obtain **9b** as white solid (method A: 74 mg; 48%; method B: 105.6 mg, 69%; method C: 53.4 mg, 38%; m.p. = 221–225 °C) and **9f** (method C: 31.2 mg, 31%; m.p. = 184–185 °C).

**9b**: ^1^H-NMR (300 MHz, DMSO*-d*_6_) (*δ*/ppm): 8.18 (4H, s, H2; H8), 7.85 (2H, s, H5′), 7.53 (4H, d, *J* = 8.7 Hz, Ph), 7.04 (4H, d, *J* = 8.7 Hz, Ph), 5.12 (4H, s, CH_2_), 4.88 (4H, t, *J* = 5.7 Hz, CH_2_CH_2_), 4.67 (4H, t, *J* = 5.7 Hz, CH_2_CH_2_), 4.16 (8H, bs, CH_2_), 1.73–1.48 (12H, m, CH_2_). ^13^C-NMR (151 MHz, DMSO*-d*_6_) (*δ*/ppm): 157.1 (Ph-q), 153.0 (C6), 151.8 (C2/C8), 150.5 (C4), 142.9 (C4′), 132.6 (Ph-q), 127.2 (Ph), 124.7 (C5′), 118.7 (C5), 115.1 (Ph), 61.1 (CH_2_), 48.6 (CH_2_), 43.0 (CH_2_), 25.6 (CH_2_), 24.2 (CH_2_). Anal. calcd. for C_42_H_46_N_16_O_2_: C, 62.52; H, 5.75; N, 27.77. Found: C, 62.62; H, 5.71; N, 27.76.

**9f**: ^1^H-NMR (300 MHz, DMSO*-d_6_*) (*δ*/ppm): 8.18 (1H, s, H2), 8.16 (1H, s, H8), 7.83 (1H, s, H5′), 7.55 (4H, dd, *J* = 8.7, 6.4 Hz, Ph), 7.05 (4H, dd, *J* = 8.8, 1.9 Hz, Ph), 5.12 (2H, s, CH_2_), 4.88 (2H, t, *J* = 5.7 Hz, CH_2_CH_2_), 4.82 (2H, d, *J* = 2.3 Hz, CH_2_CCH), 4.66 (2H, t, *J* = 5.8 Hz, CH_2_CH_2_), 4.16 (4H, bs, CH_2_), 3.58 (1H, t, *J* = 2.3 Hz, CH_2_CCH), 1.73–1.50 (6H, m, CH_2_). ^13^C-NMR (75 MHz, DMSO*-d*_6_) (*δ*/ppm): 157.3 (Ph-q), 156.4 (Ph-q), 153.2 (C6), 152.0 (C2), 150.6 (C4), 143.0 (C4′), 139.4 (C8), 133.1 (Ph-q), 132.6 (Ph-q), 127.4 (Ph), 127.3 (Ph), 124.8 (C5′), 118.8 (C5), 115.3 (Ph), 115.2 (Ph), 79.4 (CCH), 78.3 (CCH), 61.2 (CH_2_), 55.5 (CH_2_), 48.7 (CH_2_), 43.1 (CH_2_), 25.7 (CH_2_), 24.3 (CH_2_). Anal. calcd. for C_30_H_30_N_8_O_2_: C, 67.40; H, 5.66; N, 20.96. Found: C, 67.38; H, 5.41; N, 20.99.

1,3-Bis{1-[2-(6-(piperidin-1-yl)-9H-purin-9-yl)ethyl]-1H-1,2,3-triazol-4-yl}propane (**9c**) and 9-{2-[4-(pent-4-yn-1-yl)-1H-1,2,3-triazol-1-yl]ethyl}-6-(piperidin-1-yl)-9H-purine (**9g**). Compounds **9c** and **9g** were prepared according to the above-mentioned procedure from 1,6-heptadiyne **4c** (0.06 mL, 0.54 mmol) and compound **3c** (324.1 mg, 1.19 mmol) to obtain **9c** as white solid (method A: 137.3 mg, 40%; method B: 214.3 mg, 62%; method C: 78.4 mg, 21%, m.p. = 218–220 °C) and **9g** (method C: 20.7 mg, 10%; m.p. = 171–174 °C).

**9c**: ^1^H-NMR (300 MHz, DMSO*-d*_6_) (*δ*/ppm): 8.17 (2H, s, H2), 7.81 (2H. s, H8), 7.72 (2H, s, H5′), 4.80 (4H, t, *J* = 5.6 Hz, CH_2_CH_2_), 4.62 (4H, t, *J* = 5.7 Hz, CH_2_CH_2_), 4.14 (4H, s, CH_2_), 1.86–1.69 (2H, m, CH_2_CH_2_CH_2_), 1.68–1.50 (12H, m, CH_2_). ^13^C-NMR (151 MHz, DMSO*-d*_6_) (*δ*/ppm): 153.0 (C6), 151.8 (C2), 150.5 (C4), 146.5 (C4′), 139.3 (C8), 122.2 (C5′), 118.7 (C5), 48.4 (CH_2_), 43.1 (CH_2_), 28.7 (CH_2_), 25.6 (CH_2_), 24.2 (CH_2_), 24.1 (CH_2_). Anal. calcd. for C_31_H_40_N_16_: C, 58.47; H, 6.33; N, 35.19. Found: C, 58.47; H, 6.25; N, 35.22.

**9g**: ^1^H-NMR (300 MHz, DMSO*-d*_6_) (*δ*/ppm): 8.18 (1H, s, H2), 7.81 (1H, s, H8), 7.72 (1H, s, H5′), 4.86–4.72 (2H, m, CH_2_CH_2_), 4.68–4.57 (2H, m, CH_2_CH_2_), 4.16 (4H, bs, CH_2_), 2.78 (1H, t, *J* = 2.6 Hz, CCH), 2.62 (2H, t, *J* = 7.5 Hz, CH_2_CH_2_CH_2_), 2.12 (2H, td, *J* = 7.0, 2.6 Hz, CH_2_CH_2_CH_2_), 1.77–1.43 (8H, m, CH_2_). ^13^C-NMR (151 MHz, DMSO*-d*_6_) (*δ*/ppm): 153.0 (C6), 151.8 (C2), 150.4 (C4), 146.0 (C4′), 139.2 (C8), 122.1 (C5′), 118.7 (C5), 84.0 (CCH), 71.3 (CCH), 48.4 (CH_2_), 43.0 (CH_2_), 27.7 (CH_2_), 25.5 (CH_2_), 24.2 (CH_2_), 23.8 (CH_2_), 17.0 (CH_2_). Anal. calcd. for C_19_H_24_N_8_: C, 62.62; H, 6.64; N, 30.75. Found: C, 58.47; H, 6.25; N, 35.22.

*{1-[2-(6-(Piperidin-1-yl)-9H-purin-9-yl)ethyl]-1H-1,2,3-triazol-4-yl}methyl ether* (**9d**) Compound **9d** was prepared according to the abovementioned procedure for method B from propargyl ether **4d** (0.05 mL, 0.53 mmol) and compound 3c (301.2 mg, 1.17 mmol) to obtain **9d** as yellow solid (method A: 187.3 mg, 55%; method C: 156.1 mg, 46%; m.p. = 107–109 °C). ^1^H-NMR (600 MHz, DMSO*-d_6_*) (*δ*/ppm): 8.17 (2H, s, H2), 8.03 (2H, s, H8), 7.82 (2H, s, H5′), 4.88–4.84 (4H, m, CH_2_CH_2_), 4.67–4.64 (4H, m, CH_2_CH_2_), 4.42 (4H, s, CH_2_OCH_2_), 4.13 (8H, bs, CH_2_), 1.67–1.62 (4H, m, CH_2_), 1.57–1.51 (8H, m, CH_2_). ^13^C-NMR (75 MHz, DMSO*-d_6_*) (*δ*/ppm): 153.1 (C6), 151.9 (C2), 150.5 (C4), 142.1 (C4′), 139.3 (C8), 126.3 (C5′), 118.8 (C5), 62.2 (CH_2_), 48.6 (CH_2_), 43.1 (CH_2_), 25.7 (CH_2_), 24.2 (CH_2_). Anal. calcd. for C_30_H_38_N_16_O: C, 56.41; H, 6.00; N, 35.09. Found: C, 56.56; H, 6.03; N, 34.98.

*1,4-Bis{[1-(2-(6-(pyrrolidin-1-yl)-9H-purin-9-yl)ethyl)-1H-1,2,3-triazol-4-yl]methoxy}benzene* (**10a**). Compound **10a** was prepared according to the abovementioned procedure from compound **4a** (50 mg, 0.27 mmol) and compound **3d** (152.8 mg, 0.59 mmol) to obtain **10a** as white solid (method A: 53.2 mg, 28%; method B: 100.7 mg, 61%; method C: 99 mg; 60%, m.p. = 253–256 °C). ^1^H-NMR (300 MHz, DMSO*-d*_6_) (*δ*/ppm): 8.17 (2H, s, H2), 8.09 (2H, s, H8), 7.79 (2H, s, H5′), 6.90 (4H, s, Ph), 5.00 (4H, s, OCH_2_), 4.94–4.76 (4H, m, CH_2_CH_2_), 4.76–4.59 (4H, m, CH_2_CH_2_), 4.01 (4H, bs, CH_2_), 3.62 (4H, bs, CH_2_), 1.93 (8H, bs, CH_2_). ^13^C-NMR (151 MHz, DMSO*-d*_6_) (*δ*/ppm): 152.4 (Ph-q), 152.3 (C6), 152.2 (C2), 149.9 (C4), 143.0 (C4′), 124.5 (C5′), 119.2 (C5), 115.7 (Ph), 61.6 (CH_2_), 48.6 (CH_2_), 42.9 (CH_2_). Anal. calcd. for C_34_H_38_N_16_O: C, 58.11; H, 5.45; N, 31.89. Found: C, 58.15; H, 5.55; N, 32.00.

4,4′-Bis{[1-(2-(6-(piperidin-1-yl)-9H-purin-9-yl)ethyl)-1H-1,2,3-triazol-4-yl]methoxy}-1,1′-biphenyl (**10b**) and 6-(piperidin-1-yl)-9-{2-[4-(((4′-(prop-2-yn-1-yloxy)-[1,1′-biphenyl]-4-yl)oxy)methyl)-1H-1,2,3-triazol-1-yl]ethyl}-9H-purine (**10f**). Compounds **10b** and **10f** were prepared according to the abovementioned procedure from compound **4b** (50 mg, 0.19 mmol) and compound **3d** (107.3 mg, 0.42 mmol) to obtain **10b** as white solid (method A: 88.4 mg, 60%; method B: 118.4 mg, 80%; method C: 71.5 mg, 49%; m.p. = 259–262 °C) and **10f** as yellow solid (method C: 60.5 mg, 61%; m.p. = 134–137 °C).

**10b**: ^1^H-NMR (300 MHz, DMSO*-d*_6_) (*δ*/ppm): 8.17 (2H, s, H2), 8.14 (2H, s, H8), 7.80 (2H, s, H5′), 7.53 (4H, d, *J* = 8.7 Hz, Ph), 7.04 (4H, d, *J* = 8.7 Hz, Ph), 5.12 (4H, s, OCH_2_), 4.92–4.85 (4H, m, CH_2_CH_2_), 4.70–4.63 (4H, m, CH_2_CH_2_), 4.00 (4H, bs, CH_2_), 3.61 (4H, bs, CH_2_), 1.92 (8H, bs, CH_2_). ^13^C-NMR (151 MHz, DMSO*-d*_6_) (*δ*/ppm): 157.1 (Ph-q), 152.4 (C6), 152.1 (C2), 149.8 (C4), 142.8 (C4′), 136.3 (C8), 132.6 (Ph-q), 127.1 (Ph), 124.5 (C5′), 119.1 (C5), 115.1 (Ph), 61.2 (CH_2_), 48.5 (CH_2_), 42.8 (CH_2_), 41.4 (CH_2_). Anal. calcd. for C_42_H_46_N_16_O_2_: C, 62.52; H, 5.75; N, 27.77. Found: C, 62.51; H, 5.81; N, 27.96.

**10f**: ^1^H-NMR (300 MHz, DMSO*-d*_6_) (*δ*/ppm): 8.17 (1H, s, H2), 8.14 (1H, s, H8), 7.80 (1H, s, H5′), 7.55 (4H, dd, *J* = 8.7, 7.1 Hz, Ph), 7.04 (4H, d, *J* = 8.7 Hz, Ph), 5.12 (2H, s, CH_2_), 4.91–4.86 (2H, m, CH_2_CH_2_), 4.83 (2H, d, *J* = 2.3 Hz, CH_2_CCH), 4.70–4.64 (m, CH_2_CH_2_), 3.69–3.53 (3H, m, CH_2_, CCH), 1.93 (4H, s, CH_2_). ^13^C-NMR (151 MHz, DMSO*-d*_6_) (*δ*/ppm): 157.2 (Ph-q), 156.4 (Ph-q), 152.5 (C6), 152.3 (C2), 149.9 (C4), 146.89, 143.0 (C4′), 139.8 (C8), 133.1 (Ph-q), 132.6 (Ph-q), 127.4 (Ph), 127.3 (Ph), 124.7 (C5′), 119.3 (C5), 115.3 (Ph), 115.3 (Ph), 79.4 (CCH), 78.2 (CCH), 61.2 (CH_2_), 55.5 (CH_2_), 48.7 (CH_2_), 43.0 (CH_2_). Anal. calcd. for C_30_H_30_N_8_O: C, 67.40; H, 5.66; N, 20.96. Found: C, 67.27; H, 5.71; N, 23.14.

*1,3-Bis{1-[2-(6-(piperidin-1-yl)-9H-purin-9-yl)ethyl]-1H-1,2,3-triazol-4-yl}propane* (**10c**). Compound **10c** was prepared according to the abovementioned procedure from 1,6-heptadiyne **4c** (50 mg, 0.54 mmol) and compound **3d** (305.7 mg, 1.19 mmol) to obtain **10c** as white solid (method A: 163.5 mg, 50%; method B: 150.8 mg, 82%; method C: 196.2 mg, 58%; m.p. = 217–220 °C).

**10c**: ^1^H-NMR (600 MHz, DMSO*-d*_6_) (*δ*/ppm): 8.15 (1H, s, H2), 7.76 (1H, s, H8), 7.69 (1H, s, H5′), 4.90–4.69 (4H, m, CH_2_CH_2_), 4.75–44.52 (4H, m, CH_2_CH_2_), 3.98 (4H, bs, CH_2_), 3.58 (4H, bs, CH_2_), 2.48 (m, CH_2_CH_2_CH_2_), 1.90 (8H, bs, CH_2_), 1.83–1.70 (2H, m, CH_2_CH_2_CH_2_). ^13^C-NMR (75 MHz, DMSO*-d*_6_) (*δ*/ppm): 152.4 (C6), 152.2 (C2), 149.9 (C4), 146.4 (C4′), 139.8 (C8), 122.2 (C5′), 119.2 (C5), 48.4 (CH_2_), 43.0 (CH_2_), 28.7 (CH_2_), 24.1 (CH_2_). Anal. calcd. for C_31_H_40_N_16_: C, 58.47; H, 6.33; N, 35.19. Found: C, 58.22; H, 6.19; N, 34.99.

*{1-[2-(6-(Piperidin-1-yl)-9H-purin-9-yl)ethyl]-1H-1,2,3-triazol-4-yl}methyl ether* (**10d**) Compound **10d** was prepared according to the abovementioned procedure for method B from **4d** (50 mg, 0.53 mmol) and compound **3d** (301.3 mg, 1.17 mmol) to obtain **10d** as yellow solid (method A: 164.2 mg, 48%; method C: 197 mg, 58%; m.p. = 112–115 °C). ^1^H-NMR (300 MHz, DMSO*-d*_6_) (*δ*/ppm): 8.16 (2H, s, H2), 8.05 (2H, s, H8), 7.81 (2H, s, H5′), 4.86 (4H, s, CH_2_), 4.66 (4H, s, CH_2_), 4.40 (4H, s, CH_2_), 3.97 (4H, s, CH_2_), 3.59 (4H, s, CH_2_), 1.91 (8H, s CH_2_). ^13^C-NMR (151 MHz, DMSO*-d*_6_) (*δ*/ppm): 152.5 (C6), 152.2 (C2), 149.9 (C4), 143.8 (C4′), 139.5 (C8), 124.4 (C5′), 119.2 (C5), 62.2 (CH_2_OCH_2_), 48.7 (CH_2_), 43.0 (CH_2_). Anal. calcd. for C_30_H_38_N_16_O: C, 56.41; H, 6.00; N, 35.09. Found: C, 56.78; H, 5.89; N, 35.02.

1,4-Bis{[1-(2-(6-amino-9H-purin-9-yl)ethyl)-1H-1,2,3-triazol-4-yl]methoxy}benzene (**11a**) and 6-amino-9-{2-[4-((4-(prop-2-yn-1-yloxy)phenoxy)methyl)-1H-1,2,3-triazol-1-yl]ethyl}-9H-purine (**11e**). Compounds **11a** and **11e** were prepared according to the above-mentioned procedure from compound **4a** (100 mg, 0.49 mmol) and compound **3e** (200.7 mg, 1.08 mmol) to obtain **11a** as white solid (method B: 28.9 mg, 11%; m.p. > 280 °C) and **11e** as white solid (method C: 47.4 mg, 25%; m.p. > 250 °C).

**11a**: ^1^H-NMR (600 MHz, DMSO*-d*_6_) (*δ*/ppm): 8.11 (2H, s, H2), 8.09 (2H, s, H8), 7.80 (2H, s, H5′), 7.20 (4H, s, NH_2_), 6.90 (4H, s, Ph), 5.01 (4H, s, CH_2_), 4.87 (4H, t, *J* = 5.8 Hz, CH_2_CH_2_), 4.65 (4H, t, *J* = 5.8 Hz, CH_2_ CH_2_). ^13^C-NMR (151 MHz, DMSO*-d*_6_) (*δ*/ppm): 155.6 (Ph-q), 152.2 (C6), 152.1 (C2), 149.3 (C4), 142.9 (C4′), 124.0 (C5′), 118.5 (C5), 115.7 (Ph), 61.7 (CH_2_), 48.3 (CH_2_), 42.6 (CH_2_). Anal. calcd. for C_26_H_26_N_16_O_2_: C, 52.52; H, 4.41; N, 37.69. Found: C, 52.38; H, 4.17; N, 37.91.

**11e**: ^1^H-NMR (300 MHz, DMSO*-d*_6_) (*δ*/ppm): 8.11 (2H, s, Hz, H2; H8), 7.80 (1H, s, H5′), 7.20 (2H, s, NH_2_), 6.92 (4H, d, *J* = 1.6 Hz, Ph), 5.01 (2H, s, CH_2_), 4.87 (2H, t, *J* = 5.7 Hz, CH_2_CH_2_), 4.72 (2H, d, *J* = 2.3 Hz, CH_2_CCH), 4.65 (2H, t, *J* = 5.7 Hz, CH_2_CH_2_), 3.53 (t, *J* = 2.3 Hz, CH_2_CCH). ^13^C-NMR (75 MHz, DMSO*-d*_6_) (*δ*/ppm): 155.9 (Ph-q), 152.6 (C6), 152.5 (C2), 151.4 (Ph-q), 149.5 (C4), 143.0 (C4′), 140.5 (C8), 124.6 (C5′), 118.6 (C5), 115.8 (Ph), 115.6 (Ph), 79.5 (CCH), 78.0 (CCH), 6.5 (CH_2_), 55.9 (CH_2_), 48.6 (CH_2_), 43.0 (CH_2_). Anal. calcd. for C_19_H_18_N_8_O_2_: C, 58.45; H, 4.65; N, 28.70. Found: C, 58.30; H, 4.75; N, 28.87.

*6-Amino-9-{2-[4-(pent-4-yn-1-yl)-1H-1,2,3-triazol-1-yl]ethyl}-9H-purine* (**11g**) Compound **11g** was prepared according to the abovementioned procedure from 1,6-heptadiyne **4c** (50 mg, 0.53 mmol) and compound **3e** (243.7 mg, 1.17 mmol) to obtain **11g** as white solid (method C: 89.3 mg, 55%; m.p. = 226–228 °C). ^1^H-NMR (600 MHz, DMSO*-d*_6_) (*δ*/ppm): 8.11 (1H, s, H2), 7.78 (1H, s, H8), 7.74 (1H, s, H5′), 7.17 (2H, s, NH_2_), 4.79 (2H, t, *J* = 5.8 Hz, CH_2_CH_2_), 4.61 (2H, t, *J* = 5.8 Hz, CH_2_CH_2_), 2.77 (1H, t, *J* = 2.6 Hz, CCH), 2.62 (2H, t, *J* = 7.5 Hz, CH_2_), 2.14 (2H, td, *J* = 7.1, 2.6 Hz, CH_2_), 1.72–1.65 (2H, m, CH_2_). ^13^C-NMR (151 MHz, DMSO*-d*_6_) (*δ*/ppm): 155.9 (C6), 152.4 (C2), 149.4 (C4), 146.0 (C4′), 140.4 (C8), 122.1 (C5′), 118.5 (C5), 84.0 (CCH), 71.4 (CCH), 48.5 (CH_2_), 42.9 (CH_2_), 27.8 (CH_2_), 23.8 (CH_2_), 17.0 (CH_2_). Anal. calcd. for C_14_H_16_N_8_: C, 56.74; H, 5.44; N, 37.81. Found: C, 56.64; H, 5.51; N, 38.00.

{1-[2-(6-Amino-9H-purin-9-yl)ethyl]-1H-1,2,3-triazol-4-yl}methyl ether (**11d**) and 6-amino-9-{2-[4-((prop-2-yn-1-yloxy)methyl)-1H-1,2,3-triazol-1-yl]ethyl}-9H-purine (**11h**). Compounds **11d** and **11h** were prepared according to the abovementioned procedure for method B from propargyl ether **4d** (100 mg, 1.06 mmol) and compound **3e** (432.9 mg, 2.12 mmol) to obtain **11d** as white solid (method C: 48.9 mg, 8%, m.p. = 191–193 °C) and **11h** as white solid (method C: 20.8 mg, 7%, m.p. = 175–180 °C).

**11d**: ^1^H-NMR (600 MHz, DMSO*-d*_6_) (*δ*/ppm): 8.17 (2H, s, H2), 8.02 (2H, s, H8), 7.89 (2H, s, H5′), 7.57 (4H, bs, NH_2_), 4.86 (4H, t, *J* = 5.7 Hz, CH_2_CH_2_), 4.66 (4H, t, *J* = 5.7 Hz, CH_2_CH_2_), 4.50 (4H, s, CH_2_OCH_2_). ^13^C-NMR (151 MHz, DMSO*-d*_6_) (*δ*/ppm): 155.8 (C6), 152.4 (C2), 149.4 (C4), 143.6 (C4′), 124.1 (C5′), 118.5 (C5), 62.1 (CH_2_), 48.5 (CH_2_), 42.9 (CH_2_). Anal. calcd. for C_20_H_22_N_16_O: C, 47.80; H, 4.41; N, 44.60. Found: C, 47.81; H, 4.46; N, 44.59.

**11h**: ^1^H-NMR (300 MHz, DMSO*-d*_6_) (*δ*/ppm): 8.11 (1H, s, H2), 8.02 (1H, s, H8), 7.80 (1H, s, H5′), 7.20 (2H, s, NH_2_), 4.85 (2H, t, *J* = 5.7 Hz, CH_2_CH_2_), 4.64 (2H, t, *J* = 5.7 Hz, CH_2_CH_2_), 4.50 (2H, s, OCH_2_), 4.09 (2H, d, *J* = 2.4 Hz, CH_2_CCH), 3.46 (1H, t, *J* = 2.4 Hz, CH_2_CCH). ^13^C-NMR (151 MHz, DMSO*-d*_6_) (*δ*/ppm): 155.9 (C6), 152.4 (C2), 149.5 (C4), 143.2 (C4′), 140.4 (C8), 124.4 (C5′), 118.6 (C5), 79.9 (CCH), 77.4 (CCH), 61.8 (CH_2_), 56.3 (CH_2_), 48.6 (CH_2_), 43.0 (CH_2_). Anal. calcd. for C_13_H_14_N_8_O: C, 52.34; H, 4.73; N, 37.56. Found: C, 52.47; H, 4.69; N, 37.36.

### 3.6. General Procedure for the Synthesis of Bis-Pyrrolo[2,3-d]pyrimidine Derivatives **6a**–**6****d** and **7a**–**7d**

The corresponding bis-pyrrolo[2,3-*d*]pyrimidine **5a**–**5d** and cyclic amine (4 eq) were dissolved in water (3 mL). The reaction mixture was stirred under microwave irradiation (400 W) at 100 °C during 10 min. The reaction mixture was triturated with acetonitrile to obtain the crude product.

*1,4-Bis{[1-(2-(4-(piperidin-1-yl)-7H-pyrrolo[2,3-d]pyrimidin-7-yl)ethyl)-1H-1,2,3-triazol-4-yl]methoxy}benzene* (**6a**). Compound **6a** was prepared according to the abovementioned procedure from compound **5a** (100 mg, 0.16 mmol) and piperidine (0.06 mL, 0.64 mmol) to obtain **6a** as white solid (68.9 mg, 59%, m.p. = 185–187 °C). ^1^H-NMR (600 MHz, DMSO*-d*_6_) (*δ*/ppm): 8.13 (2H, s, H2), 8.01 (2H, s, H5′), 6.97–6.78 (6H, m, Ph; H6), 6.48 (2H, d, *J* = 3.7 Hz, H5), 5.01 (4H, s, OCH_2_), 4.80 (4H, t, *J* = 5.9 Hz, CH_2_CH_2_), 4.62 (4H, t, *J* = 5.9 Hz, CH_2_CH_2_), 3.92–3.74 (8H, m, CH_2_), 1.69–1.50 (12H, m, CH_2_). ^13^C-NMR (75 MHz, DMSO*-d*_6_) (*δ*/ppm): 156.2 (Ph-q), 152.2 (C4), 150.7 (C7a), 150.7 (C2), 142.9 (C4′), 124.5 (C5′), 123.9 (C6), 115.6 (Ph), 102.1 (C4a), 100.9 (C5), 61.8 (OCH_2_), 49.0 (CH_2_CH_2_), 46.2 (CH_2_CH_2_), 43.9 (CH_2_), 25.4 (CH_2_), 24.2 (CH_2_). Anal. calcd. for C_38_H_44_N_14_O_2_: C, 62.62; H, 6.09; N, 26.90. Found: C, 62.79; H, 5.92; N, 27.15.

*4,4′-Bis{[1-(2-(4-(piperidin-1-yl)-7H-pyrrolo[2,3-d]pyrimidin-7-yl)ethyl)-1H-1,2,3-triazol-4-yl]methoxy}-1,1′-biphenyl* (**6b**). Compound **6b** was prepared according to the abovementioned procedure from compound **5b** (70 mg, 0.09 mmol) and piperidine (0.04 mL, 0.37 mmol) to obtain **6b** as white solid (51.9 mg, 73%, m.p. = 137–139 °C). ^1^H-NMR (600 MHz, DMSO*-d*_6_) (*δ*/ppm): 8.13 (1H, s, H2), 8.06 (1H, s, H5′), 7.53 (4H, d, *J* = 8.7 Hz, Ph), 7.04 (4H, d, *J* = 8.8 Hz, Ph), 6.88 (2H, d, *J* = 3.6 Hz, H6) 6.45 (2H, d, *J* = 3.7 Hz, H5), 5.12 (4H, s, OCH_2_), 4.81 (4H, t, *J* = 5.9 Hz, CH_2_CH_2_), 4.63 (4H, t, *J* = 5.9 Hz, CH_2_CH_2_), 3.82–3.79 (8H, m, CH_2_), 1.78– 1.45 (12H, m, CH_2_). ^13^C-NMR (75 MHz, DMSO*-d*_6_) (*δ*/ppm): 157.1 (Ph-q), 156.2 (C4), 150.7 (C7a), 150.7 (C2), 142.8 (C4′), 132.5 (Ph-q), 127.2 (Ph), 124.6 (C5), 123.9 (C5′), 115.2 (Ph), 102.1 (C4a), 100.9 (C6), 61.1 (OCH_2_), 49.1 (CH_2_CH_2_), 46.2 (CH_2_CH_2_), 43.9 (CH_2_), 25.4 (CH_2_), 24.2 (CH_2_). Anal. calcd. for C_44_H_48_N_14_O_2_: C, 65.65; H, 6.01; N, 24.36. Found: C, 65.89; H, 6.11; N, 24.23.

*1,3-Bis{1-[2-(4-(piperidin-1-yl)-7H-pyrrolo[2,3-d]pyrimidin-7-yl)ethyl]-1H-1,2,3-triazol-4-yl}propane* (**6c**). Compound **6c** was prepared according to the abovementioned procedure from compound **5c** (50 mg, 0.09 mmol) and piperidine (0.04 mL, 0.37 mmol) to obtain **6c** as white solid (51.3 mg, 87%, m.p. = 145–147 °C). ^1^H-NMR (600 MHz, DMSO*-d*_6_) (*δ*/ppm): 8.11 (2H, s, H2), 7.57 (2H, s, H5′), 6.92 (2H, d, *J* = 3.7 Hz, H6), 6.50 (2H, d, *J* = 3.7 Hz, H5), 4.73 (4H, t, *J* = 5.9 Hz, CH_2_CH_2_), 4.60 (4H, t, *J* = 5.9 Hz, CH_2_CH_2_), 3.98–3.64 (8H, m, CH_2_), 2.47 (8H, t, *J* = 7.4 Hz, CH_2_CH_2_CH_2_), 1.76–1.69 (2H, m, CH_2_CH_2_CH_2_), 1.66–1.60 (4H, m, CH_2_), 1.56–1.50 (8H, m, CH_2_). ^13^C-NMR (151 MHz, DMSO*-d*_6_) (*δ*/ppm): 156.3 (C4), 150.8 (C7a), 150.7 (C2), 146.4 (C4′), 124.0 (C5), 122.2 (C5′), 102.2 (C4a), 101.0 (C6), 49.0 (CH_2_CH_2_), 46.3 (CH_2_CH_2_), 44.1 (CH_2_), 28.7 (CH_2_), 25.5 (CH_2_), 24.2 (CH_2_), 24.1 (CH_2_). Anal. calcd. for C_33_H_42_N_14_: C, 62.44; H, 6.67; N, 30.89. Found: C, 62.38; H, 6.91; N, 30.97.

*{1-[2-(4-(Piperidin-1-yl)-7H-pyrrolo[2,3-d]pyrimidin-7-yl)ethyl]-1H-1,2,3-triazol-4-yl}methyl ether* (**6d**). Compound **6d** was prepared according to the abovementioned procedure from compound **5d** (50 mg, 0.09 mmol) and piperidine (0.04 mL, 0.37 mmol) to obtain **6d** as white solid (34.7 mg, 59%, m.p. = 141–143 °C). ^1^H-NMR (300 MHz, DMSO*-d_6_*) (*δ*/ppm): 8.12 (2H, s, H2), 7.90 (2H, s, H5′), 6.95 (2H, d, *J* = 3.1 Hz, H6), 6.51 (2H, d, *J* = 3.1 Hz, H5), 4.84–4.75 (4H, m, CH_2_CH_2_), 4.66–4.58 (4H, m, CH_2_CH_2_), 4.38 (4H, s, CH_2_OCH_2_), 3.92–3.66 (8H, m, CH_2_), 1.74–1.43 (12H, m, CH_2_). ^13^C-NMR (75 MHz, DMSO*-d_6_*) (*δ*/ppm): 156.1 (C4), 150.7 (C7a), 150.7 (C2), 143.5 (C4′), 124.2 (C5′), 123.9 (C6), 102.1 (C4a), 100.9 (C5), 62.0 (OCH_2_), 49.0 (CH_2_CH_2_), 46.2 (CH_2_CH_2_), 44.0 (CH_2_), 25.4 (CH_2_), 24.2 (CH_2_). Anal. calcd. for C_32_H_40_N_14_O: C, 60.36; H, 6.33; N, 30.80. Found: C, 60.51; H, 6.57; N, 30.88.

*1,4-Bis{[1-(2-(4-(pyrrolidin-1-yl)-7H-pyrrolo[2,3-d]pyrimidin-7-yl)ethyl)-1H-1,2,3-triazol-4-yl]methoxy}benzene* (**7a**). Compound **7a** was prepared according to the abovementioned procedure from compound **5a** (50 mg, 0.10 mmol) and pyrrolidine (0.04 mL, 0.40 mmol) to obtain **7a** as white solid (46.3 mg, 65%, m.p. = 193–195 °C). ^1^H-NMR (300 MHz, DMSO*-d*_6_) (*δ*/ppm): 8.09 (2H, s, H2), 8.00 (2H, s, H5′), 6.89 (4H, s, Ph), 6.84 (2H, d, *J* = 3.5 Hz, H6), 6.48 (2H, d, *J* = 3.6 Hz, H5), 5.00 (4H, s, OCH_2_), 4.80 (4H, t, *J* = 5.6 Hz, CH_2_CH_2_), 4.62 (4H, t, *J* = 5.6 Hz, CH_2_CH_2_), 3.67 (8H, bs, CH_2_), 1.93 (8H, bs, CH_2_). ^13^C-NMR (75 MHz, DMSO*-d*_6_) (*δ*/ppm): 154.7 (C4), 152.2 (C7a), 151.2 (C2), 149.8 (Ph-q), 142.9 (C4′), 124.5 (C5′), 123.4 (C6), 115.7 (Ph), 102.6 (C4a), 100.7 (C5), 61.5 (OCH_2_), 49.0 (CH_2_), 47.4 (CH_2_), 43.8 (CH_2_). Anal. calcd. for C_36_H_40_N_14_O_2_: C, 61.70; H, 5.75; N, 27.98. Found: C, 61.52; H, 5.90; N, 28.01.

*4,4′-Bis{[1-(2-(4-(pyrrolidin-1-yl)-7H-pyrrolo[2,3-d]pyrimidin-7-yl)ethyl)-1H-1,2,3-triazol-4-yl]methoxy}-1,1′-biphenyl* (**7b**). Compound **7b** was prepared according to the above-mentioned procedure from compound **5b** (70 mg, 0.10 mmol) and pyrrolidine (0.04 mL, 0.40 mmol) to obtain **7b** as white solid (46.3 mg, 6%, m.p. = 187–189 °C). ^1^H-NMR (600 MHz, DMSO*-d*_6_) (*δ*/ppm): 8.09 (2H, s, H2), 8.04 (2H, s, H5′), 7.53 (4H, d, *J* = 8.7 Hz, Ph), 7.04 (4H, d, *J* = 8.8 Hz, Ph), 6.82 (2H, d, *J* = 3.5 Hz, H5), 6.45 (2H, d, *J* = 3.6 Hz, H6), 5.12 (4H, s, OCH_2_), 4.80 (4H, t, *J* = 5.9 Hz, CH_2_CH_2_), 4.62 (4H, t, *J* = 5.9 Hz, CH_2_CH_2_), 3.65 (8H, bs, CH_2_), 1.92 (8H, bs, CH_2_). ^13^C-NMR (75 MHz, DMSO*-d_6_*) (*δ*/ppm): 157.1 (Ph-q), 154.6 (C4), 151.1 (C2), 149.7 (C7a), 142.8 (C4′), 132.5 (Ph-q), 127.2 (Ph), 124.6 (C6), 123.4 (C5′), 115.2 (Ph), 102.6 (C4a), 100.8 (C5), 61.1 (OCH_2_), 49.1 (CH_2_), 47.4 (CH_2_), 43.8 (CH_2_). Anal. calcd. for C_42_H_44_N_14_O_2_: C, 64.93; H, 5.71; N, 25.24. Found: C, 64.92; H, 5.78; N, 25.12.

*1,3-Bis{1-[2-(4-(pyrrolidin-1-yl)-7H-pyrrolo[2,3-d]pyrimidin-7-yl)ethyl]-1H-1,2,3-triazol-4-yl}propane* (**7c**). Compound **7c** was prepared according to the abovementioned procedure from compound **5c** (50 mg, 0.09 mmol) and pyrrolidine (0.04 mL, 0.37 mmol) to obtain **7c** as white solid (53.1 mg, 97%, m.p. = 170–172 °C). ^1^H-NMR (300 MHz, DMSO*-d*_6_) (*δ*/ppm): 8.07 (2H, s, H2), 7.54 (2H, s, H5′), 6.87 (2H, d, *J* = 3.5 Hz, H6), 6.48 (2H, d, *J* = 3.6 Hz, H5), 4.73 (4H, t, *J* = 5.7 Hz, CH_2_CH_2_), 4.59 (4H, t, *J* = 5.6 Hz, CH_2_CH_2_), 3.64 (8H, bs, CH_2_), 2.46 (4H, t, *J* = 7.4 Hz, CH_2_CH_2_CH_2_), 1.91 (8H, bs, CH_2_), 1.79–1.62 (2H, m, CH_2_CH_2_CH_2_). ^13^C-NMR (151 MHz, DMSO*-d*_6_) (*δ*/ppm): 154.6 (C4), 151.1 (C2), 149.7 (C7a), 146.2 (C4′), 123.3 (C6), 122.0 (C5′), 102.6 (C4a), 100.7 (C5), 48.8 (CH_2_), 47.3 (CH_2_), 43.9 (CH_2_), 28.6 (CH_2_), 24.0 (CH_2_). Anal. calcd. for C_31_H_38_N_14_: C, 61.37; H, 6.31; N, 32.32. Found: C, 61.32; H, 6.71; N, 32.10.

*{1-[2-(4-(Pyrrolidin-1-yl)-7H-pyrrolo[2,3-d]pyrimidin-7-yl)ethyl]-1H-1,2,3-triazol-4-yl}methyl ether* (**7d**). Compound **7d** was prepared according to the abovementioned procedure from compound **5d** (50 mg, 0.09 mmol) and pyrrolidine (0.04 mL, 0.37 mmol) to obtain **7d** as white solid (39.5 mg, 65%, m.p. = 131–135 °C). ^1^H-NMR (300 MHz, DMSO*-d*_6_) (*δ*/ppm): 8.07 (2H, s, H2), 7.85 (2H, s, H5′), 6.89 (2H, d, *J* = 2.2 Hz, H6), 6.50 (2H, d, *J* = 2.6 Hz, H5), 4.84–4.73 (4H, m, CH_2_CH_2_), 4.69–4.56 (4H, m, CH_2_CH_2_), 4.36 (4H, s, CH_2_OCH_2_), 3.64 (8H, bs, CH_2_), 1.92 (8H, bs, CH_2_). ^13^C-NMR (151 MHz, DMSO*-d*_6_) (*δ*/ppm): 154.7 (C4), 151.2 (C2), 149.8 (C7a), 143.4 (C4′), 124.1 (C5′), 123.3 (C6), 102.6 (C4a), 100.8 (C5), 62.0 (CH_2_), 49.0 (CH_2_), 47.4 (CH_2_), 43.8 (CH_2_). Anal. calcd. for C_30_H_36_N_14_O: C, 59.20; H, 5.96; N, 32.22. Found: C, 59.03; H, 5.90; N, 37.56.

### 3.7. Computational Details

All molecular geometries were optimized using the B97D functional together with the 6–31+G(d) basis set for non-metals and the Stuttgart-Dresden (SDD) effective core potentials [61] for the inner electrons of copper atoms and its associated double-ζ basis set for the outer ones, in line with our earlier work on the copper-catalyzed organic reactions [62] and other literature recommendations [63,64]. To account for the solvent effects, during geometry optimization, we included the implicit SMD solvation model corresponding to DMF (ε = 37.219). Thermal corrections were extracted from the matching frequency calculations, so that all presented results correspond to differences in the Gibbs free energies at room temperature and normal pressure. The choice of such computational setup was prompted by its success in reproducing various features of different organic [65,66,67], organometallic [68,69] and biological systems [70,71], being particularly accurate for relative trends among similar reactants, which is the focus here. All transition state structures were located using the scan procedure, employing both 1D and 2D scans, the latter specifically utilized to probe the possibility for concerted mechanisms. Apart from the visualization of the obtained negative frequencies, the validity of all transition state structures was validated by performing IRC calculations in both directions and identifying the matching reactant and product structures connected by the inspected transition state. All calculations were performed using the Gaussian 16 software [72].

### 3.8. Cell Culturing

Human carcinoma cell lines A549 (lung carcinoma), HeLa (cervical carcinoma), SW620 (colorectal adenocarcinoma, metastatic) and CFPAC-1 (pancreatic cancer, derived from metastatic: liver), and normal human foreskin (HFF-1) fibroblasts were obtained from the American Type Culture Collection (ATCC, Manassas, VA, USA). Cells were cultured in humidified atmosphere at 37 °C with 5% CO_2_. As growth medium, Dulbecco′s modified Eagle medium (DMEM) was used with the addition of fetal bovine serum (10%), L-glutamine (2 mM) and antibiotics: streptomycin (100 mg/mL) and penicillin (100 U/mL).

### 3.9. Proliferation Assay

Cells were seeded onto 96-well microtiter plates at a seeding density of 3000 cells/well for carcinoma cell lines, and 5000 cells/well for normal human fibroblasts. The next day, cells were treated with test agents in five different concentrations (0.01–100 µM) and further incubated for 72 h. DMSO (solvent) was tested for potential cytotoxic effect but it did not exceed 0.1%. Following 72 h incubation, the MTT assay was performed and measured absorbances were transformed into percentage of cell growth as described previously [73]. Results were obtained from three independent experiments. IC_50_ values were calculated using linear regression analysis and results were statistically analyzed by ANOVA, Tukey post-hoc test (*p* < 0.05).

### 3.10. Apoptosis Detection

Cells were seeded into 8-well chambers (Lab-tek II Chamber Slides, Thermo Fisher Scientific, Waltham, MA, USA) in a concentration of 2 × 10^4^ cells per well and treated with 2 × IC_50_ and 5 × IC_50_ concentrations of selected compounds for 48 and 72 h. Staining of the cells was performed by Annexin-V-FITC Staining kit (Santa Cruz Biotechnology, Dallas, TX, USA) according to the manufacturer′s instructions. Cells were visualised by fluorescent microscope (Olympus, Tokyo, Japan) at magnification of 40×.

## 4. Conclusions

The novel bis- (**5a**–**5d**, **6a**–**6d**, **7a**–**7d**, **9a**–**9d**, **10a**–**10d**, **11a**–**11d**) and mono-(**5f**, **5g**, **9e**−**9g**, **10f**, **10g**, **11e**, **11g** and **11h**) pyrrolo[2,3-*d*]pyrimidines and purines were synthesized. Aromatic and aliphatic spacers were introduced between *N*-heterocycles by the CuAAC reaction using different catalysts and reaction conditions (methods A–C). Non-conventional ultrasonic energy implemented in method B proved to be better in terms of selectivity, reaction time and efficiency. Ultrasound-assisted reactions led to the synthesis of exclusively symmetrical bis-heterocycles with improved yield after 1.5 h. Direct utilization of a Cu(I) source using the CuI/DIPEA/HOAc catalytic system resulted in the formation of both bis- and mono-heterocycles in most cases.

DFT calculations confirmed that the investigated copper-catalyzed cycloaddition is a feasible process, and revealed that triazoles are favorably formed in a concerted and highly exergonic fashion, Δ*G*_R_ = −51.0 kcal mol^−1^ for the studied model case. Still, the uncatalyzed reaction is associated with a high kinetic barrier of Δ*G*^‡^ = 28.5 kcal mol^−1^, which renders it as very unlikely. Once Cu(I) ions are present, they bind to the alkyne and increase its electrophylicity towards the azide, which reduces kinetic requirements to Δ*G*^‡^ = 19.7 kcal mol^−1^, thereby allowing the conversion to occur under normal conditions. The obtained reaction profiles agree that higher temperatures and ultrasound irradiation will improve the reaction outcomes, while eliminate the option to use strong bases with the prospect to activate alkynes through terminal C–H deprotonation. Lastly, different trends among reactions where catalytic Cu(I) ions are generated in situ or directly introduced could be ascribed to the presence of Cu(II) ions in the former, but this depends on their availability and other conditions that might be operative when a large variety of reagents is employed as was the instance here. Lastly, a somewhat general tendency of 6-amino derivatives to offer lower reaction yields over 6-chloro analogues is likely related to their ability to more strongly bind Cu(I) in solution, therefore hindering its catalytic efficiency.

Antiproliferative evaluations showed that the linker between the heterocyclic scaffolds had a significant impact on antitumor activity. Thus, cyclic amino-substituted bis-pyrrolo[2,3-*d*]pyrimidines **7a** and purines **9b** and **10b** with aromatic 1,4-bis(oxymethylene)phenyl and 4,4′-bis(oxymethylene)biphenyl spacer exhibited potent cytostatic activity, especially on HeLa cell lines (**7a**: IC_50_ = 6.4 µM; **9b**: IC_50_ = 3.8 µM; **10b**: IC_50_ = 7.4 µM). Among the mono-heterocyclic derivatives, compounds **5f**, **9f** and **10f** with 4,4′-bis(oxymethylene)biphenyl unit at C-4 of 1,2,3-triazole showed the most potent antitumor activity. The 4-chloropyrrolo[2,3-*d*]pyrimidine **5f** showed the most pronounced inhibitory effect against HeLa (IC_50_ = 0.98 µM) and CFPAC-1 (IC_50_ = 0.79 µM) cell lines. Conversely, the majority of bis-heterocycles with aliphatic central unit had marginal activity or were devoid of any cytostatic activity. Among heterocycles, 4-chloropyrrolo[2,3-*d*]pyrimidine and 6-piperidinylpurine made a major contribution to the antiproliferative effect. Compound **5f** was further investigated by Annexin V assay, which showed that its growth-inhibitory effect on CFPAC-1 could be ascribed to the induction of apoptosis and primary necrosis.

Our findings encourage further structural optimization of purine and fused heterocycle scaffolds, such as chloropyrrolo[2,3-*d*]pyrimidine connected through aromatic unit, as a promising chemical entity for development of novel efficient and non-toxic agent against pancreatic cancer.

## Data Availability

Appendix A with 1H and 13C NMR spectra and Appendix A containing data for computational analyses are available on line.

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
