# Peer review of "Novel Bis- and Mono-Pyrrolo[2,3-d]pyrimidine and Purine Derivatives: Synthesis, Computational Analysis and Antiproliferative Evaluation"

_molecules, 2021, doi:10.3390/molecules26113334_

Round 1

Reviewer 1 Report

The article presented “Novel bis- and mono-pyrrolo[2,3-d]pyrimidine and purine derivatives: synthesis, computational analysis and antiproliferative evaluation” aims at developing and studying a class of “clicked” symmetrical modified nucleobases for cancer treatment. The authors synthetized a good array of novel compounds and evaluated their efficacy against four cancer cell lines, and confronted the activity with relevant reference compounds. Three different methodologies were evaluated for the click reaction optimization. In addition, computational studies regarding the CuAAC mechanism were conducted.

Overall, the work is well structured and presented clearly to the reader, the article is recommended for publication with minor revisions.

Author Response

The article presented “Novel bis- and mono-pyrrolo[2,3-d]pyrimidine and purine derivatives: synthesis, computational analysis and antiproliferative evaluation” aims at developing and studying a class of “clicked” symmetrical modified nucleobases for cancer treatment. The authors synthetized a good array of novel compounds and evaluated their efficacy against four cancer cell lines, and confronted the activity with relevant reference compounds. Three different methodologies were evaluated for the click reaction optimization. In addition, computational studies regarding the CuAAC mechanism were conducted. Overall, the work is well structured and presented clearly to the reader, the article is recommended for publication with minor revisions.

Minor revisions:

  1. Line 121: correct “that found” with “that was found”.

It is now corrected (p. 4).

  1. Line 151 (170,177): correct “CUAAC” with “CuAAC”, unify in all text.

The abbreviation CuAAC is adjusted throughout the manuscript.

  1. Lines 153-155: needs clarification. In the table product 5f is obtained with method C and 5g is obtained with method A and C. In text is reported that 5g and 5f are obtained with method B and C.

We thank the reviewer to this comment. The text is now corrected (p. 5).

  1. Lines 167-168: the authors claim that the ultrasonic irradiation (method B) decreased reaction time and afforded better yields when compared to classical conditions (methods A, C). However, when method B is applied the reaction is conducted at 80 °C while methods A and C envisage reactions at room temperature. Is it possible that the observed effects are due to the difference in reaction temperature of the methods since no experiment is conducted at room temperature under ultrasound irradiation? Authors should add a comment to better explain.

The reactions under ultrasound irradiation (method B) caused cavitation and streams within the bath that instantly increased the temperature of ultrasonic bath to 80 °C. Optimizations of conditions in classical methods A and C using different catalytic systems suggested that these reactions are more efficient when conducted at room temperature. Namely, when the classical reactions (methods A and C) were carried out at 80 °C, additional by-products were obtained.

  1. Lines 180-183: needs rephrasing. The meaning of the sentence is not very clear, maybe a sentence explaining that 3e could not react with 4b in any condition could help clarify the matter

We thank the reviewer for suggestion and have made this change to clarify the sentence (p. 7).

  1. Lines 234-237: there are no rate constants in the text or in the supporting information.

The reviewer is right as no rate constants are indeed measured. Yet, the referred part of the discussion concerns the estimate of the catalytic effect of Cu(I) that is done based on the computationally modelled reduction in the kinetic barrier of 8.8 kcal/mol for the copper-catalyzed reaction. Accordingly, the later value would translate into 6-7 orders of magnitude higher rate constant, and this is how it is presented in the text.

  1. Line 261: correct “the undesired way” with “an undesired way”.

It has been corrected (p. 9).

  1. Line 352: correct “what is in accordance” with “that is in accordance”.

It has been changed as suggested (p. 11).

  1. Line 360: correct “had the significant impact” with “had a significant impact”.

It has been corrected (p. 11).

  1. Line 536: correct the number of protons for the signal at 7.03 ppm, (4 H d J=8.7 Hz)

The number of protons has been corrected (p. 16).

  1. Line 574: compound 5d was not synthetized with method B as mentioned in the text. Correct with methods A,C

Compound 5d was obtained using methods A and C that is now corrected (p. 17). We apologize for this mistake.

  1. Lines 780-783: “The corresponding bis-pyrrolo[2,3-d]pyrimidine (5a–5d) and cyclic amine (4 eq) were 780 dissolved in water (3 ml). The reaction mixture was stirred under microwave irradiation 781 (400 W) at 100 ˚C during 10 min. The reaction mixture was washed with acetonitrile to 782 obtain the crude product”. What does “washed with acetonitrile” means in this case?

It means that the reaction mixture was triturated with acetonitrile to obtain the crude product. This has been changed in general procedure for the synthesis (p. 21).

  1. Figure S46 (lines 303-314): Comparison is made between 6-chloro pyrrolo[2,3-d]pyrimidine derivatives and 6-amino purine derivatives regarding the complexation of Cu(I) ions. The comparison would be more relevant if made between 6-chloro and 6-ammino purines or 6-chloro and 6-ammino pyrrolo[2,3- d]pyrimidines?

The reviewer is fully right and we have extended Figure S46 (in Supplementary information) including now the appropriate 6-chloro and 6-amino derivatives of both pyrrolo[2,3-d]pyrimidines and purines. This did not change any conclusions as amino derivatives, in all cases, are more strongly binding Cu(I) in solution and this holds consistently for both classes of organic skeletons. Therefore, we have also inserted a comment on page 10.

Reviewer 2 Report

The submitted manuscript is of a very high quality. The Authors have synthesized a series of compounds using unusual method of synthesis and evaluated their biological activity in multiple appropriate tests. The manuscript is not only scientifically sound but also the results are nicely presented and the discussion is comprehensive. My comments are minor ones.

Introduction, the structures of the compounds discussed in lines 79-83 should be presented.

Figure 1, from this figure it is very hard to find what are the structures of the studied compounds. I appreciate such a figure, but I think it should be redesigned, maybe by adding some tags such as “5a” next to the chosen groups or linkers directly in the figure? Right now it is simply not very informative.

Figure S41, m1-, Cu2+, why the position of Cu ion is significantly different than in the above two? Were the initial orientations (with Cu0 and Cu+) the same as in the case of Cu2+?

Line 879, B97D is quite unusual, why this functional has been chosen? And don’t you think that B97D3 would be a better choice?

The basis set is rather small, I have hoped it would be 6-311++, the molecules are not very large.

In the computational part the Authors discuss the possibility of using Cu(0) as a catalyst. Of course, everything can be calculated, but how can you imagine such conditions in the real experiment? Solubility of Cu(0) in DMF is very low so that even if the this metal would be formed in situ it would precipitate.

In order to suggest the mechanism of action of the newly synthesized compounds, haven’t you considered docking to one (or many) of the proteins listed in lines 65-67 and 73-74?

Author Response

The submitted manuscript is of a very high quality. The Authors have synthesized a series of compounds using unusual method of synthesis and evaluated their biological activity in multiple appropriate tests. The manuscript is not only scientifically sound but also the results are nicely presented and the discussion is comprehensive.

We are thankful to this reviewer for his/her encouraging words.

My comments are minor ones.

  1. Introduction, the structures of the compounds discussed in lines 79-83 should be presented.

The structures of presented compounds are now included in Figure 1.

  1. Figure 1, from this figure it is very hard to find what are the structures of the studied compounds. I appreciate such a figure, but I think it should be redesigned, maybe by adding some tags such as “5a” next to the chosen groups or linkers directly in the figure? Right now it is simply not very informative.

Figure 1 has been changed by adding the compound numbers that are assigned to the corresponding structural motifs. We hope that the figure is now more informative.

  1. Figure S41, m1-, Cu2+, why the position of Cu ion is significantly different than in the above two? Were the initial orientations (with Cu0 and Cu+) the same as in the case of Cu2+?

The reviewer is right and we are very thankful for bringing this aspect to our attention. Indeed, all of our calculations were performed by allowing several initial orientations for the considered metals and each organic ligand, and data in Figure S41 correspond to the most stable complexes. However, the initial version of Figure S41 contained a wrong picture for the complex between Cu+ and m1, which appeared confusing. We have re-checked our calculations and inserted the right picture, which now shows that the most stable complexes between deprotonated alkyne m1 and copper are all linear regardless of the metal's oxidation state.

  1. Line 879, B97D is quite unusual, why this functional has been chosen? And don’t you think that B97D3 would be a better choice?

The basis set is rather small, I have hoped it would be 6-311++, the molecules are not very large.

We agree with the reviewer that increasing the size of the basis set and changing the DFT functional from B97D to B97D3 might formally appear as a good strategy to improve the quality and reliability of computational results. However, although such a systematical benchmark study for copper is, to the best of our knowledge, not available in the literature, in selecting our computational approach we relied on two things: (1) in our experience, the employed (SMD)/B97D/6–31+G(d)/SDD level of theory already performed well for copper-catalyzed organic reactions (reference [62] in the text), and (2) the use of B97D functional for copper has been either recommended or employed by others as well (references [63,64] in the text). This is now clarified in the paragraph 3.7. Computational details (p. 23).

  1. In the computational part the Authors discuss the possibility of using Cu(0) as a catalyst. Of course, everything can be calculated, but how can you imagine such conditions in the real experiment? Solubility of Cu(0) in DMF is very low so that even if the this metal would be formed in situ it would precipitate.

Besides using direct utilization of a copper(I) source and alternative creation of copper(I) through the reduction of a copper(II) source, the CuAAC reactions can be catalyzed by Cu(I) species supplied by elemental copper, thus further simplifying the experimental procedure. The copper metal procedure is facile protocol particularly convenient and used for high-throughput synthesis of compound libraries for biological screening. Cu(II) sulfate may be added to accelerate the reaction; however, this is not necessary in most cases, as copper oxides and carbonates, the patina on the metal surface, are sufficient to initiate the catalytic cycle. Moreover, finding in our earlier work (Beilstein J. Org. Chem. 13, 2017, 2352) and by others (Chem. Soc. Rev. 39, 2010, 1302) demonstrated the ability to conduct CuAAC reactions simply my milling process using elemental copper vial or copper balls that strongly suggested the reaction was occurred on the surface of the vial or ball.

  1. In order to suggest the mechanism of action of the newly synthesized compounds, haven’t you considered docking to one (or many) of the proteins listed in lines 65-67 and 73-74?

This is certainly a good suggestion and worth considering. Still, we feel that a full computational account of this aspect would certainly extend much beyond the scope of the current manuscript. We believe this requires a separate study and will be addressed in our future work together with several other classes of organic ligands prepared and experimentally evaluated in our team.

Author Response

In this manuscript the authors report the synthesis and antiproliferative studies for a series of bis-purines and bis-7-deazapurines. In addition to the biochemical evaluation, computational studies are reported that focus on the mechanism and energetics for formation of the bis-heterocycles. The authors vary the linker and the substituents on the purine and 7-deazapurines and identify an efficient method for the synthesis of the series of compounds using ultrasound. While some compounds did not show observable inhibitory activity in the experiments conducted, several compounds inhibited growth of normal fibroblasts and the tumor cell lines investigated. Overall, this manuscript will advance understanding of this class of compounds as inhibitors and should be of interest for the readership of the journal. However, before accepting for publication a few comments are offered for consideration.

  1. The authors comment that the linker had an effect on the antitumor activity as shown in Table 2, but it would strengthen the manuscript if additional description of the results for inhibition assays could be provided. Most compounds are not as effective as the previously published compounds used for comparison (except for 5f and 9f). It would be helpful to provide some analysis as why many compounds may not be effective inhibitors and others may show inhibition similar to the previously published compounds.

Reviewer is right that most compounds from presented class are not as effective as the previously published structural analogues BIS-PP1, BIS-PP2. However, from a series of 31 prepared compounds, we succeeded to obtain compound 5f with better antiproliferative activity on HeLa and CFPAC-1 cells compared to BIS-PP1, BIS-PP2. The aim of the presented study was to prepare novel bis-pyrrolo[2,3-d]pyrimidine and bis-purine analogues and evaluate their antiproliferative activity to find candidates with selective and strong antitumor activity. The 5f was highlighted as a candidate, therefore, its ability to induce apoptosis was further evaluated. Importantly, this compound showed to be more selective than BIS-PP1, BIS-PP2 exhibiting lower toxicity to normal fibroblasts (HFF-1), what is now emphasized in the paragraph 2.1.1. Antiproliferative evaluations on page 11.

Insights into structure-activity relationship that are obtained from the results of antiproliferative evaluations are given in the text and depicted in the Figure 5. We may speculate that kinetic solubility, lipophilicity, permeability and metabolic stability of presented compounds is related with their antiproliferative activity. However, for more detailed analyses, evaluations of physico-chemical and ADME properties need to be carried out, that would be beyond the scope of the current manuscript. In the next phase of our study, lead optimization process, we certainly plan to perform extensive in vitro physicochemical and ADME property investigations of the representatives of this class, along with other selected nitrogen-containing heterocycles that exhibited promising antiproliferative activity.

  1. The authors can comment on the substituents chosen for this study. The previous study is referenced, but it would help if there were some information as to why a chloro or tertiary amine are evaluated as the substituents (and how the starting adenine-based compound is not an effective inhibitor).

The reviewer is right at this point. Design strategy in development of purine derivatives as cytostatic agents for kinase inhibition revealed that introduction of pyrrolidine and piperidine ring improved the activity by forming an additional hydrogen bond to kinase hinge (Med. Chem. Res. 2018, 27, 1384; RSC Med. Chem. 2020, 11, 1112). In addition, prevalence of halogenated drugs showed that halogenated compounds have been widely exploited in drug discovery confirming that the formation of halogen bonds contribute to the stability of protein-ligand complexes (J. Med. Chem. 2013, 56, 1363.). Encouraged with this findings, structural diversity was extended to introduction of cyclic amines and the synthesis of halogen-substituted bis-pyrrolo[2,3-d]pyrimidines and bis-purines that were presented in this work. These findings, along with references, were now included in the paragraph 1. Introduction (p. 2).

  1. In the conclusions, it is stated that the DFT calculations revealed the reaction is a feasible process (line 936), but this was demonstrated in generating the product. It is suspected that feasible is referring to the reaction being exergonic. The authors provide great detail regarding the energetics of the reaction and how the energetic barrier is reduced in the presence of copper ions, but overall they are defining a catalyst. It would help to better clarify the role of the computational studies in evaluating the potential structural roles of the catalyst in the reaction. Many comparisons are made in the supporting info, but the overall contribution of this part of the study may be lost in the details presented.

The reviewer is right and we are thankful for acknowledging a range of useful information and insights offered through computational analysis. To address this remark, we have slightly broadened the Conclusions section by introducing several comments. In addition, when we refer in the text to a reaction being feasible, we mean not only it is thermodynamically exergonic, but also that its kinetic barrier is such to allow the transformation to proceed under normal conditions. In this respect, we have shown that the uncatalyzed reaction is associated with an extensive barrier of ΔG = 28.5 kcal mol–1, yet the presence of Cu(I) ions reduces it to ΔG = 19.7 kcal mol–1, thereby allowing the product formation under employed conditions. We have improved the discussion along these lines at several places in the text.

  1. There are a number of grammatical errors throughout the manuscript. Examples:

Line 79 – not indented

The line is now indented.

  1. Line 122 – missing article before CuAAC

The reference is provided.

  1. Line 304 – comma not needed for dependent clause

The comma is omitted.

  1. Line 305 ‘analogously compared’ can be more clear if stated as ‘compared’

The change has been done as suggested.

  1. Minor points

-The abbreviation for CuAAC is not used consistently (sometimes CUAAC).

The abbreviations for CuAAC have been unified throughout the manuscript.

  1. The CuAAC abbreviation is defined on lines 134-135 but is used prior to the definition.

The CuAAC abbreviation is defined when the first time CuACC is mentioned in the texts (in Abstract).

  1. In the methods, the number of significant figures varies for the mass values (some given to the hundredths and others to the tenths, for example lines 576 versus 586).

Decimal place values for mass given in hundredths are rounded to tenths.

  1. If the 13C NMR data are DEPT data, this can be noted in the experimental (noted based on negative peaks shown in the spectra).

The 13C NMR spectra are actually attached proton test (APT) spectra, that is now noted in the Experimental.